# Genetic determinants of daytime napping and effects on cardiometabolic health

Hassan S. Dashti [1,2,3,14], Iyas Daghlas[1,2,14], Jacqueline M. Lane[1,2,3], Yunru Huang[4], Miriam S. Udler [1,2,5,6], Heming Wang [2,7], Hanna M. Ollila[1,2,8,9], Samuel E. Jones [8,10], Jaegil Kim[11], Andrew R. Wood[10], 23andMe Research Team*, Michael N. Weedon[10], Stella Aslibekyan[4], Marta Garaulet [7,12,13,15] & Richa Saxena [1,2,3,15]

Daytime napping is a common, heritable behavior, but its genetic basis and causal relationship with cardiometabolic health remain unclear. Here, we perform a genome-wide association study of self-reported daytime napping in the UK Biobank ($n = 452,633$) and identify 123 loci of which 61 replicate in the 23andMe research cohort ($n = 541,333$). Findings include missense variants in established drug targets for sleep disorders (HCRTR1, HCRTR2), genes with roles in arousal (TRPC6, PNOC), and genes suggesting an obesity-hypersomnolence pathway (PNOC, PATJ). Association signals are concordant with accelerometer-measured daytime inactivity duration and 33 loci colocalize with loci for other sleep phenotypes. Cluster analysis identifies three distinct clusters of nap-promoting mechanisms with heterogeneous associations with cardiometabolic outcomes. Mendelian randomization shows potential causal links between more frequent daytime napping and higher blood pressure and waist circumference.

[1] Center for Genomic Medicine, Massachusetts General Hospital and Harvard Medical School, Boston, MA, USA. [2] Broad Institute, Cambridge, MA, USA. [3] Department of Anesthesia, Critical Care and Pain Medicine, Massachusetts General Hospital and Harvard Medical School, Boston, MA, USA. [4] 23andMe, Inc., Sunnyvale, CA, USA. [5] Diabetes Unit, Massachusetts General Hospital, Boston, MA, USA. [6] Department of Medicine, Harvard Medical School, Boston, MA, USA. [7] Division of Sleep and Circadian Disorders, Brigham and Women's Hospital and Harvard Medical School, Boston, MA, USA. [8] Institute for Molecular Medicine FIMM, HiLIFE, University of Helsinki, Helsinki, Finland. [9] Department of Psychiatry and Behavioral Sciences, Stanford University, Stanford, CA, USA. [10] Genetics of Complex Traits, University of Exeter Medical School, Exeter, UK. [11] GlaxoSmithKline, Waltham, MA, USA. [12] Department of Physiology, University of Murcia, Murcia, Spain. [13] IMIB-Arrixaca, Murcia, Spain. [14] These authors contributed equally: Hassan S. Dashti, Iyas Daghlas. [15] These authors jointly supervised this work: Marta Garaulet, Richa Saxena. *A list of authors and their affiliations appears at the end of the paper. ✉email: garaulet@um.es; rsaxena@mgh.harvard.edu

Naps are short daytime sleep episodes that are evolutionarily conserved across diverse diurnal species ranging from flies[1] to polyphasic mammals[2]. In human adults, daytime napping is highly prevalent in Mediterranean cultures and is also common in non-Mediterranean countries including the United States[3]. In modern society, napping is encouraged in sleep-deprived populations, such as night shift workers[4] and airline pilots[5], to acutely improve performance and alertness. Although an acute benefit of napping on increased arousal in the setting of sleep deprivation is well-established[6], the long-term effects of habitual napping on chronic disease risk remain controversial. Indeed, cross-sectional studies have provided conflicting evidence on the effects of habitual napping on cognition, blood pressure, obesity, metabolic traits, and mortality[7–13]. As napping behavior may be confounded by inadequate nighttime sleep or underlying poor health[14,15], causal inference from these observational studies is limited.

Genetic variation constitutes an important contributor to interindividual differences in napping preference. A twin study estimated heritability of self-reported napping and objective daytime sleep duration to be 65% and 61%, respectively, demonstrating heritability similar or even higher than heritability found for other sleep traits such as nighttime sleep duration and timing[16]. Indeed, up to seven genetic loci for daytime napping have been discovered in genome-wide association study (GWAS) of self-reported napping or related accelerometer-derived sleep measures[17–19]. Discovery of additional genetic loci may reveal biological pathways regulating sleep, elucidate genetic links with other sleep and metabolic traits, and clarify the potential causal effects of habitual napping on cardiometabolic disease.

In this work, we leverage the full UK Biobank dataset of European ancestry, including related individuals ($n = 452,633$), and an independent replication sample from 23andMe research participants of European ancestry ($n = 541,333$), to define the genetic architecture of daytime napping and to assess links with other sleep and cardiometabolic traits. We identify 123 loci of which 61 replicate in the 23andMe research cohort, including variants in established drug targets for sleep disorders (*HCRTR1*, *HCRTR2*), genes with roles in arousal (*TRPC6*, *PNOC*), and genes suggesting an obesity-hypersomnolence pathway (*PNOC*, *PATJ*). Cluster analysis identifies three distinct clusters of nap-promoting mechanisms and Mendelian randomization shows potential causal links between more frequent daytime napping and higher blood pressure and waist circumference.

## Results

Among UK Biobank participants of European ancestry ($n = 452,633$), 38.2% and 5.3% of participants reported sometimes and always napping, respectively (Supplementary Table 1). Participants reporting always napping were more likely to be older males, report longer 24 h sleep duration and more frequent daytime sleepiness, have higher body-mass index (BMI), waist circumference, systolic and diastolic blood pressures, have diagnosed sleep apnea, have a higher Townsend deprivation index (i.e., greater degree of socio-economic deprivation), and report being current smokers, unemployed or retired, and shift workers (all $P < 0.001$; Supplementary Table 1).

### Discovery, validation, and replication of 123 genetic loci for daytime napping in UK Biobank and 23andMe. We conducted GWAS using 13,304,133 high-quality imputed genetic variants across 452,633 participants. We identified 123 distinct loci, with ($P < 5 \times 10^{-8}$; Fig. 1A, Supplementary Data 1, Supplementary Fig. 1a) genome-wide SNP-based heritability estimated at 11.9% (standard error = 0.1%). The 123 loci explained 1.1% of the

variance in daytime napping. The LD score regression intercept was 1.04 and therefore did not indicate uncontrolled confounding. Effect estimates were largely consistent in GWAS restricted to 338,764 participants self-reporting excellent or good overall health (Supplementary Table 1, Supplementary Data 1). As higher BMI is associated with more frequent napping[20], we conducted a GWAS adjusting for BMI alone or BMI and BMI × BMI and found that 110 of the 123 loci retained genome-wide significance (Supplementary Data 1). Accounting for sleep apnea in GWAS models excluding participants with diagnosed sleep apnea ($n = 5553$ excluded) or adjusting by a modified STOP-BANG risk scale[21] did not influence findings (Supplementary Data 1). Finally, when adjusting for daytime sleepiness, we observed modest attenuation of effect estimates, with 60 of the 123 loci retaining genome-wide significance (Supplementary Data 1).

We found no evidence of sexual dimorphism in the autosomal genetic determinants of daytime napping behavior[22] as indicated by the lack of statistical heterogeneity by sex at any of the lead loci (all $P > 0.005$) (Supplementary Data 1) and a genome-wide genetic correlation ($r_g$) of male and female stratified GWAS of 0.94 (standard error = 0.03). We conducted association analyses on the X chromosome to further examine whether common variants on the X chromosome contribute to sex differences in daytime napping and identified five additional loci for daytime napping (Supplementary Table 2). Only one of these variants (rs6621715) had significantly different effect estimates in males and females ($P = 0.006$), and no additional GWAS signals were identified on the X chromosome in sex-stratified analysis.

Five of seven loci for daytime napping reported in earlier GWAS in a subset of unrelated UK Biobank participants of European ancestry ($n = 386,577$)[18] retained genome-wide significance in our analyses (Supplementary Table 3). However, none of the suggestive loci reported in GWAS of accelerometer-derived phenotypes related to napping behavior in the UK Biobank ($n = 85,670$)[19] and LIFE Adult Study ($n = 956$)[17] showed evidence of association in the current analysis.

We tested for independent replication of lead loci using data from 23andMe, Inc., a personal genetics company, where 541,333 research participants of European ancestry also provided data on the frequency of daytime napping (43.0% sometimes and 7.6% always napping; Supplementary Table 4). We replicated 61 of 109 tested loci ($P < 4.6 \times 10^{-4}$), of which 18 of the 61 loci were genome-wide significant (i.e., $P < 5.0 \times 10^{-8}$). All 109 tested loci showed consistent direction of effect with the effect estimated in the UK Biobank ($P_{binomial} = 3.21 \times 10^{-8}$) (Fig. 1B, Supplementary Data 2). In fixed-effects inverse-variance weighted meta-analysis of UK Biobank and 23andMe (total $n = 993,966$), 94 of the 109 lead variants remained genome-wide significant (Fig. 1C, Supplementary Data 2).

Given inherent limitations of self-reported data, we aimed to partly validate the specificity of our associations with an objective measure corresponding to daytime napping behavior. We thus compared effect estimates of the 123 loci with effect estimates for accelerometer-derived daytime inactivity duration[19] from 7-day wrist accelerometry obtained in 85,499 participants of European ancestry in the UK Biobank >2 years after baseline assessment. Estimates of 90 variants were directionally concordant ($P_{binomial} = 2.74 \times 10^{-7}$) and variants at *ASCL4* and *SNAP91* were strongly associated with longer duration of daytime inactivity ($P_{adj} < 0.05$) (Supplementary Data 3). We further quantified the impact of daytime napping on daytime inactivity duration using a polygenic score comprised of lead variants at all 123 loci. A category increase in frequency of daytime napping was associated with 18.9 min (95% confidence interval = 13.6, 24.2; $P = 4.21 \times 10^{-12}$) longer duration of daytime inactivity, but had no effect on other accelerometer-derived sleep duration, timing, or quality phenotypes (Supplementary Table 5).

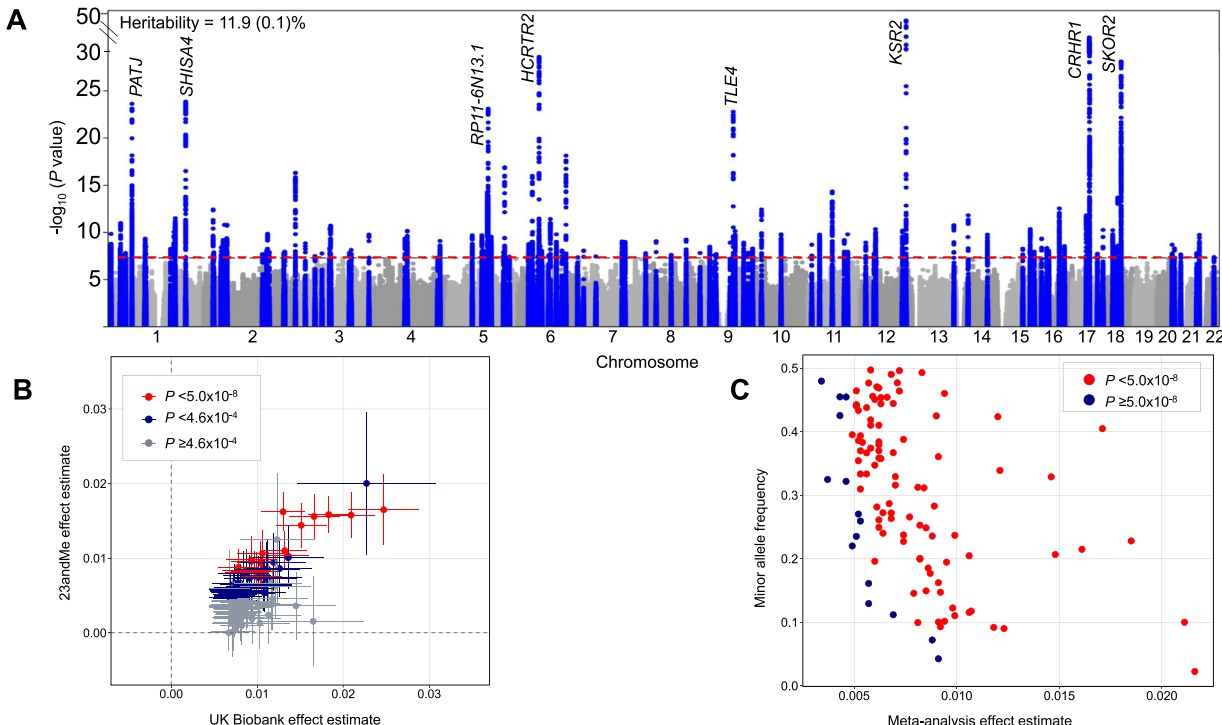

**Fig. 1 Plots for genome-wide association analysis results for daytime napping in the UK Biobank ($n = 452,633$) and replication in 23andMe ($n = 541,333$). A** Manhattan plot of daytime napping genome-wide association study in the UK Biobank ($n = 452,633$). Plot shows the $-\log_{10}P$ values (y-axis) for all genotyped and imputed single-nucleotide polymorphisms (SNPs) passing quality control (BOLT-LMM mixed-model association test $P$ values) ordered by chromosome and base position (x-axis). Blue peaks represent genome-wide significant loci. Horizontal red line denotes genome-wide significance ($P = 5 \times 10^{-8}$). Top 8 loci are annotated with nearest gene. **B** Daytime napping signals' effect estimates from UK Biobank ($n = 452,633$) plotted against effect estimates from 23andMe ($n = 541,333$). Error bars represent the 95% confidence intervals for each effect estimate. **C** Effect estimates of daytime napping signals from UK Biobank and 23andMe meta-analysis (total $n = 993,966$) plotted against minor allele frequency.

**Napping genetic variants share causal variants with other sleep phenotypes and lie near known genes that regulate arousal.** Several daytime napping-associated variants had pleiotropic associations with other self-reported sleep traits[23–26] and accelerometer-derived sleep measures[19] (Supplementary Data 3, 4). This genetic overlap between daytime napping and other sleep traits was further supported by cross-trait LD score regression[27] where we observed the strongest evidence for a shared genetic basis with daytime sleepiness ($r_g = 0.70$, $P = 7.94 \times 10^{-373}$) and long sleep duration ($r_g = 0.42$, $P = 1.94 \times 10^{-64}$), and weaker correlations with other sleep duration, timing and quality phenotypes (Supplementary Table 6). In concordance with the null polygenic risk score association, daytime napping was not genetically correlated with accelerometer-defined sleep duration. Despite the observed genome-wide genetic overlap, lead variants at 26 of the 123 loci showed no statistical evidence for association with previously studied sleep traits in the UK Biobank ($P_{adj} > 0.05$), suggesting that these variants reflect mechanisms specific to daytime napping (Supplementary Data 4).

Several genetic variants for daytime napping were located in or near genes with known effects on sleep-wake regulation. Thus, to gain insights into putative causal variants driving daytime napping and sleep-wake biology, we integrated results from functional annotation, fine-mapping, multi-trait, and eQTL colocalization analyses (for each colocalization analysis we report a posterior probability for a shared causal variant in the association signal) (Supplementary Data 5, Supplementary Tables 7–9). Functional annotation of all variants identified an enrichment of variants in intronic (46.2%) and intergenic (31.5%) regions, suggesting that non-coding gene regulatory mechanisms

may underlie napping as they do for many other complex traits (Supplementary Fig. 1b).

In order to identify association signals with evidence for shared causal variants with other sleep traits, we performed multi-trait colocalization analyses[28] of daytime napping loci across six self-reported sleep traits (daytime sleepiness, sleep duration, insomnia, snoring, chronotype, and ease of awakening) and identified 33 shared signals (of which 25 corresponded to a genome-wide significant daytime napping locus) (Supplementary Data 5). These analyses prioritized putatively causal SNPs genes at several loci which may form hypotheses for experimental follow-up.

First, missense variants were identified in components of the wake-promoting orexin/hypocretin neuropeptide signaling pathway: (i) in a transmembrane helical domain of *HCRTR2* [I308V; rs2653349; A allele frequency = 0.21; associated with more frequent daytime napping, morning preference and ease of awakening, posterior probability of colocalization (pp) = 0.98], (ii) in a cytoplasmic domain of *HCRTR1* [I408V; rs2271933, $r^2 = 0.98$ with lead rs6663012 variant; A allele frequency = 0.38; associated with more frequent daytime napping], and (iii) a cytoplasmic domain of *TRPC6* [P15S; rs3802829, $r^2 = 0.98$ with lead rs11224896 variant; G allele frequency = 0.89; associated with more napping and longer sleep duration; pp=0.80], which encodes a subunit for transient receptor channels that maintains hypocretin/orexin neurons in a depolarized state[29]. Although an intronic lead variant in *HCRTR2* was previously reported in GWAS of daytime sleepiness[24] (rs3122170, $P$ value for association with daytime napping $4.60 \times 10^{-18}$; $r^2 = 0.29$ with lead napping variant rs2653349), the traits in the colocalization cluster excluded the daytime sleepiness phenotype, suggesting that the

observed napping signal is driven by a distinct causal variant in *HCRTR2* (Supplementary Table 7). To further explore the independence of these signals, we used GCTA COJO to perform conditional analysis adjusting the regional napping associations for the lead napping signal in *HCRTR2*. We found substantial attenuation in the association with napping for the lead daytime sleepiness variant in *HCRTR2* (rs3122170; *P* value from $4.60 \times 10^{-18}$ to $4.56 \times 10^{-3}$ after conditioning). This further suggests that the identified signal for daytime napping is distinct from the previously reported daytime sleepiness signal in the *HCRTR2* region.

Second, colocalization analyses revealed variants in *PNOC* and *PATJ* with effects on napping, daytime sleepiness, and BMI, suggesting a potential obesity-hypersomnolence pathway. An intronic candidate causal variant in *PNOC* [rs351776; C allele frequency = 0.55] associated with more frequent napping, more daytime sleepiness, and higher BMI. *PNOC* encodes a preproprotein that is proteolytically processed to generate the nociceptin neuropeptide, which opposes the effects of hypocretin to reduce arousal and spontaneous activity in zebrafish[30,31]. The colocalization of daytime napping with BMI at this locus is consistent with known pleiotropic effects of *PNOC* in feeding behavior[32] (pp = 0.84; Supplementary Table 8). The known[33] missense variant in *PATJ* [rs12140153; G1543V; G allele frequency 0.90] has a stronger association with daytime napping than any previously studied sleep phenotypes (Supplementary Data 4), and is likely a shared causal variant with daytime sleepiness, chronotype, and with BMI (pp = 0.81 and pp = 0.99; Supplementary Tables 7 and 8).

Third, colocalization analyses refined genetic effects previously described at the *KSR2* locus implicated in ERK/EGFR signaling[24,34], a pathway with an established causal role in sleep regulation in *C. elegans*, *Drosophila*, and zebrafish[35,36]. This included an intronic variant in *KSR2* (rs1846644; T allele frequency = 0.60; pp = 0.91), that is associated with more frequent napping, longer sleep duration, and increased daytime sleepiness.

Fourth, several genetic variants were prioritized at or near genes (a) coding for proteins constituting or interacting with potassium channels [rs77154532 (*KCHN8*), rs10875606 (*KCTD16*)], (b) involved in glutamate transmission [rs60920123 (*GRIN2A*), rs2284015 (*CACNG2*)], and (c) previously associated with periodic leg movements[37] and restless legs syndrome[38] [rs4236060 (*BTBD9*)].

Fifth, we found evidence of association for variants in *PRRC2C*, one of three orthologs of the *Drosophila* nocte gene[39]. Nocte targets clock neurons to synchronize molecular and behavioral rhythms to temperature cycles and influences siesta sleep in flies. We observed no gene-by-season (a proxy for ambient temperature) statistical interaction at this and any other loci (Supplementary Data 1).

We performed colocalization analyses using gene expression data from the frontal cortex in the GTEx data release v7[40] (*n* = 129), the brain tissue predominantly enriched for daytime napping signals. Daytime napping variants at *FADS1* associated with increased expression of *FADS1* (pp = 0.89) and at *ECE2* associated with increased expression of *ECE2* (pp = 0.99) (Fig. 2A, B; Supplementary Table 9). Another lead variant is near *FNDC5* (rs2786547), a gene coding for irisin, a muscle-derived hormone with putative effects on expression of sleep-regulating neuropeptides[41]. We found strong evidence for colocalization of the daytime napping signal with gene expression of *FNDC5* in skeletal muscle in the GTEx data release v7[40] (*n* = 706, pp = 0.93), with higher gene expression relating to less frequent napping (Supplementary Fig. 2). This suggests a role for *FNDC5* in a sleep-regulating mechanism outside of the central nervous system.

Finally, multi-trait clustering suggested the possibility of at least three distinct pathways influencing daytime napping.

Bayesian nonnegative matrix factorization (bNMF)[42] clustering for 123 variants with 17 self-reported and accelerometer-derived sleep traits identified 3 clusters (63% of 1000 iterations) and these same 3 clusters were also present in an additional 34% of iterations with 4 clusters (Table 1, Supplementary Data 6, Supplementary Fig. 3a) reflecting (a) sleep propensity (cluster 1; 6 contributing loci with *CRHR1*, *SKOR2*, *KSR2*, *ASCL4*, *RERE*, and *ECE2*); (b) disrupted sleep (cluster 2; 5 contributing loci with *SHISA4*, *ADO*, *NRXN3*, *FNDC5*, and *GS1-259H13.13* as lead); and (c) early sleep timing (cluster 3; 9 contributing loci with *HCRTR2*, *ALG10*, *ALG10B*, *PATJ*, *BTBD9*, *MTNR1B*, *AGAP1*, *RP11-6N13.1*, and *ZBTB5* as lead loci, notably not at known core clock genes). A fourth possible cluster, obstructive sleep apnea, was observed in 34% of 1000 iterations (Supplementary Fig. 3b). Results were corroborated with findings from an alternative unsupervised hierarchical clustering method[24] (Supplementary Fig. 3c), with clusters 1 and 2 partly overlapping with previously observed clusters for daytime sleepiness[24].

**Genes at association signals are enriched in brain and GABAergic neurons, and in neural development and opioid signaling pathways**. In order to identify tissues, neuronal subtypes and annotated pathways relevant to daytime napping, we first mapped the genes near association signals and then tested for their over-representation relative to all genes in experimental genome-wide datasets. Gene-based associations for 21,761 genes mapped with Pascal[43] are listed in Supplementary Data 7; 324 genes showed association after Bonferroni correction. The identified signals were enriched for genes predominantly expressed in brain tissues, including the frontal cortex ($P = 1.18 \times 10^{-7}$) and nucleus accumbens ($P = 1.26 \times 10^{-7}$) (Fig. 3A, Supplementary Table 10). Single-cell enrichment analyses in FUMA[44] using human brain datasets (listed in Fig. 3B) showed consistent enrichment in GABAergic neurons across several brain tissues including the prefrontal cortex and midbrain. In addition, pathway enrichment analysis using MAGMA[45] and Pascal[43] indicated enrichment of genes involved in regulation of transmission across chemical synapses, neuronal system, and opioid signaling (Fig. 3C, Supplementary Data 8, 9).

**The genetic contributors to daytime napping are shared with cardiometabolic diseases**. To gain insights into shared heritability of daytime napping with other disease and behavior traits, we performed cross-trait LD score regression[27] using publicly available GWAS data for 257 traits. Modest positive correlations were observed between daytime napping and several anthropometric and cardiometabolic diseases and traits including BMI, triglycerides, and type 2 diabetes (Fig. 4A, Supplementary Data 10), of which correlations with triglycerides remained significant in the GWAS model adjusting for BMI. To further characterize shared genetic links between daytime napping and diseases in a disease-enriched and independent health system-based clinical cohort, we conducted a phenome-wide association study (PheWAS) in the Mass General Brigham Biobank (*n* = 23,561 participants of European ancestry with genetic data)[46,47]. We generated a daytime napping genome-wide polygenic score (GPS) and tested associations with 951 ICD-code based disease categories. PheWAS showed 3 Bonferroni-significant associations (18 FDR-significant), including positive associations with essential hypertension (GPS q10 vs q1 odds ratio [95% confidence interval]: 1.30 [1.13, 1.51]), obesity (GPS q10 vs q1: 1.38 [1.18, 1.62]), and chronic nonalcoholic liver disease (GPS q10 vs q1: 1.51 [1.18, 1.92]), which encompasses diagnosis codes for chronic non-specific or nonalcoholic liver disease (Fig. 4B, C, Supplementary Data 11). We also observed associations of a polygenic

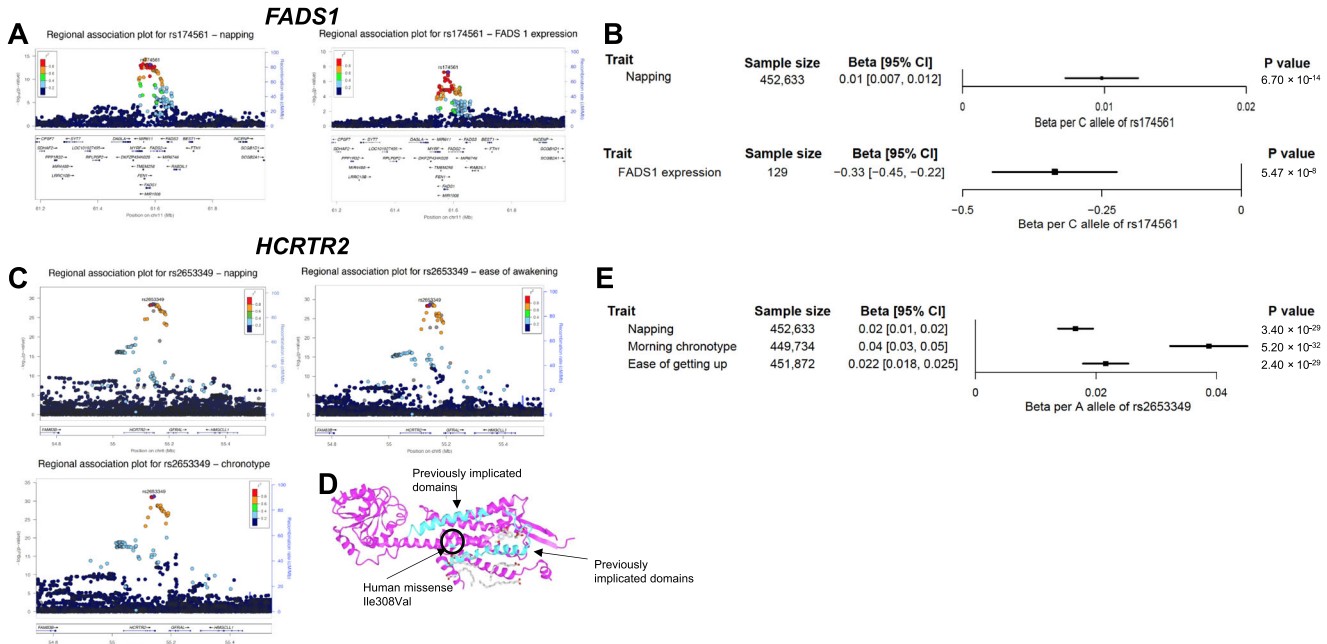

**Fig. 2 Colocalization analysis reveals a shared causal variant reducing *FADS1* gene expression in the frontal cortex and increasing napping liability, and a shared causal missense variant in *HCRTR2* influencing daytime napping, chronotype, and ease of awakening. A** Regional association plots for daytime napping and *FADS1* gene expression in the frontal cortex at rs174561 and variants within 400 kb on chromosome 11. The y-axis shows the $-\log_{10}$ *P* value for each variant in the region, and the x-axis shows the genomic position. Each variant is represented by a filled circle, with the rs174561 variant colored purple, and nearby variants colored according to degree of linkage disequilibrium ($r^2$) with rs174561. The lower panel shows genes located in the displayed region and the blue line corresponds to the recombination rate. **B** Forest plot of associations between the C allele of genetic variant rs174561 in *FADS1* with daytime napping and gene expression of *FADS1* in the frontal cortex. Units of daytime napping reflect an increase on the ordinal scale of the trait, and gene expression is in standard deviation units. *P* values are two-sided and were obtained using linear regression. Black box indicates the effect estimate and lines represent 95% confidence intervals. **C** Regional association plot for colocalized sleep phenotypes at rs2653349 and variants within 400 kb on chromosome 6. **D** Crystal structure of *HCRTR2* (PDB ID 6TPJ) showing localization of rs2653349 that changes Isoleucine to Phenylalanine or to valine at the transmembrane domain of HCRTR2. Protein sequence was visualized using iCn3D (https://www.ncbi.nlm.nih.gov/Structure/icn3d/full.html). The variant rs2653349 was aligned with the sequence (arrows to Human Missense variant in Figure) and the previously published canine *HCRTR2* mutations[105], which disrupt transmembrane and signaling domains or truncate the *HCRTR2* protein are highlighted in cyan. **E** Forest plot of associations between the A allele of genetic variant rs2653349 in *HCRTR2* and the colocalized sleep phenotypes. *P* values are two-sided and were obtained using linear regression. Black boxes show effect estimates, and surrounding lines display 95% confidence intervals.

score of the 123 napping variants, and polygenic sub-scores for each of the 3 clusters with cardiometabolic traits from large-scale public GWAS (Table 1, Supplementary Table 11). Cluster-specific polygenic score associations varied across outcomes, and included associations of cluster 1 with higher blood pressure, and clusters 2 and 3 with adiposity traits (Table 1).

**Mendelian randomization suggests a causal effect of more frequent daytime napping on increased blood pressure and waist circumference.** To explore whether daytime napping may causally increase cardiometabolic disease risk, we performed two-sample Mendelian randomization (MR) analyses using the 123 loci as genetic proxies for daytime napping (Supplementary Table 11). We observed a potentially causal effect of more frequent daytime napping on higher diastolic blood pressure (DBP; 0.25 standard deviation (SD) unit increase per category increase in daytime napping, 95% CI [0.15, 0.34], $P = 2.99 \times 10^{-7}$), systolic blood pressure (SBP; 0.18 SD units, [0.09, 0.27], $P = 5.15 \times 10^{-5}$), and waist circumference (0.28 SD units, [0.11, 0.45], $P = 1.3 \times 10^{-3}$), all of which surpassed multiple testing correction (Fig. 5A, B). In sensitivity analysis, we found a consistent effect, although attenuated in magnitude for the outcome of DBP, of genetically proxied more frequent daytime napping on higher blood pressure when using variant association statistics from 23andMe as the exposure, and blood pressure in the ICB-UKB meta-analysis[48] as the outcome (DBP: 0.08 SD units, [0.003, 1.18],

$P = 0.04$; SBP: 0.21 SD units, [−0.02, 0.43], $P = 0.07$). As the MR effects may be explained by pleiotropic effects of the napping variants on pathways independent of napping, we performed five sensitivity analyses and found consistent evidence of effect (Supplementary Data 12, Supplementary Table 12; Supplementary Fig. 4). Given prior evidence for a causal effect of higher BMI on daytime sleepiness[24], we tested the hypothesis that adiposity traits (waist circumference, waist-to-hip ratio adjusted for BMI (WHRadjBMI), and BMI) influenced daytime napping frequency. Genetically proxied WHRadjBMI was nominally associated with a modest increase in daytime napping frequency (inverse-variance weighted: 0.03 category increase in daytime napping per SD increase in WHRadjBMI, [0.01, 0.05], $P = 0.01$) (Fig. 5B, Supplementary Data 12).

**Leveraging HCRTR1 and HCRTR2 genetic associations to predict the cardiovascular safety profile of dual orexin antagonists.** Given our observation that the hypocretin pathway contributed to variation in daytime napping behavior (variants in *HCRTR1* and *HCRTR2*), and recent reports suggesting that mammalian orexin signaling has cardioprotective effects[49], we examined whether these variants may serve as instruments to predict the cardiovascular safety of orexin receptors as drug targets. This has clinical relevance, as dual orexin receptor antagonists (DORAs) are currently used as sleep medications, and orexin receptor agonists are currently in development for

**Table 1 Cluster-specific daytime napping polygenic scores associations with self-reported and accelerometer-derived sleep traits and other cardiometabolic traits.**

| Trait, units (sleep trait is a defining feature of cluster #) | Cluster 1: Sleep propensity N loci = 6 | | | Cluster 2: Disrupted sleep N loci = 5 | | | Cluster 3: Early morning awakening N loci = 9 | | |
|---|---|---|---|---|---|---|---|---|---|
| | Beta | SE | P Value | Beta | SE | P Value | Beta | SE | P Value |
| *Self-reported sleep traits* | | | | | | | | | |
| Sleep duration, minutes (1) | 0.72 | 0.07 | $1.8 \times 10^{-23}$ | −0.60 | 0.11 | $1.05 \times 10^{-07}$ | 0.14 | 0.08 | $6.77 \times 10^{-02}$ |
| Short sleep duration, log-odds (1) | −0.12 | 0.03 | $1.6 \times 10^{-04}$ | 0.34 | 0.05 | $1.54 \times 10^{-12}$ | 0.07 | 0.03 | $1.98 \times 10^{-02}$ |
| Long sleep duration, log-odds (1) | 0.19 | 0.02 | $9.0 \times 10^{-17}$ | 0.06 | 0.04 | $9.05 \times 10^{-02}$ | 0.12 | 0.02 | $6.03 \times 10^{-07}$ |
| Ease of awakening, more ease (3) | −0.18 | 0.05 | $3.3 \times 10^{-04}$ | −0.60 | 0.08 | $3.08 \times 10^{-14}$ | 0.51 | 0.05 | $4.82 \times 10^{-23}$ |
| Snoring, log-odds | −0.02 | 0.03 | $5.4 \times 10^{-01}$ | 0.14 | 0.05 | $5.14 \times 10^{-03}$ | 0.16 | 0.03 | $1.03 \times 10^{-06}$ |
| Daytime sleepiness, more sleepiness (1, 2) | 0.47 | 0.03 | $6.3 \times 10^{-47}$ | 0.48 | 0.05 | $1.76 \times 10^{-20}$ | 0.43 | 0.03 | $1.53 \times 10^{-35}$ |
| Insomnia, log-odds (2) | −0.01 | 0.04 | $7.7 \times 10^{-01}$ | 0.45 | 0.07 | $4.89 \times 10^{-11}$ | 0.11 | 0.05 | $1.27 \times 10^{-02}$ |
| Chronotype, more morningness (3) | −0.23 | 0.09 | $6.1 \times 10^{-03}$ | −0.91 | 0.13 | $8.48 \times 10^{-12}$ | 1.53 | 0.09 | $8.44 \times 10^{-66}$ |
| Obstructive sleep apnea, log-odds (3) | −0.04 | 0.10 | $6.9 \times 10^{-01}$ | 0.53 | 0.15 | $4.19 \times 10^{-04}$ | 0.53 | 0.10 | $1.00 \times 10^{-07}$ |
| *Accelerometer-derived sleep traits* | | | | | | | | | |
| Daytime inactivity duration, minutes (1, 2) | 0.85 | 0.15 | $1.4 \times 10^{-08}$ | 1.04 | 0.23 | $7.40 \times 10^{-06}$ | 0.54 | 0.15 | $4.52 \times 10^{-04}$ |
| L5 timing (midpoint of the least-active 5 h of the day), minutes (2, 3) | 0.24 | 0.15 | $1.1 \times 10^{-01}$ | 1.69 | 0.24 | $1.53 \times 10^{-12}$ | −1.33 | 0.16 | $2.36 \times 10^{-17}$ |
| M10 timing (midpoint of the most-active 10 h of the day), minutes (2, 3) | 0.16 | 0.15 | $3.1 \times 10^{-01}$ | 1.21 | 0.24 | $3.03 \times 10^{-07}$ | −1.12 | 0.16 | $8.32 \times 10^{-13}$ |
| Number of sleep bouts, n (1, 2) | −0.95 | 0.15 | $2.0 \times 10^{-10}$ | 0.45 | 0.23 | $4.93 \times 10^{-02}$ | 0.35 | 0.15 | $2.19 \times 10^{-02}$ |
| Sleep midpoint, minutes (2, 3) | 0.05 | 0.15 | $7.4 \times 10^{-01}$ | 0.88 | 0.24 | $2.65 \times 10^{-04}$ | −0.97 | 0.16 | $9.72 \times 10^{-10}$ |
| Sleep duration, minutes (1, 2) | 0.99 | 0.15 | $5.04 \times 10^{-11}$ | −0.99 | 0.23 | $2.49 \times 10^{-05}$ | 0.11 | 0.16 | $4.97 \times 10^{-01}$ |
| Sleep efficiency, % (1, 2) | 0.49 | 0.15 | $1.11 \times 10^{-03}$ | −1.41 | 0.23 | $1.59 \times 10^{-09}$ | 0.15 | 0.15 | $3.26 \times 10^{-01}$ |
| Sleep duration standard deviation, minutes | 0.23 | 0.15 | $1.42 \times 10^{-01}$ | 1.06 | 0.24 | $1.02 \times 10^{-05}$ | 0.06 | 0.16 | $7.25 \times 10^{-01}$ |
| *Cardiometabolic traits* | | | | | | | | | |
| BMI, SD kg/m² | 0.12 | 0.12 | $3.36 \times 10^{-01}$ | 1.32 | 0.22 | **$1.76 \times 10^{-09}$** | 0.37 | 0.17 | **$2.76 \times 10^{-02}$** |
| Waist circumference, SD cm | 0.09 | 0.14 | $5.43 \times 10^{-01}$ | 1.04 | 0.25 | **$2.50 \times 10^{-05}$** | 0.54 | 0.19 | **$3.92 \times 10^{-03}$** |
| Waist-hip-ratio adjusted for BMI, SD | −0.10 | 0.14 | $4.81 \times 10^{-01}$ | 0.14 | 0.23 | $5.36 \times 10^{-01}$ | 0.23 | 0.19 | $2.24 \times 10^{-01}$ |
| LDL cholesterol, SD mg/dL | 0.11 | 0.23 | $6.20 \times 10^{-01}$ | −0.11 | 0.42 | $7.99 \times 10^{-01}$ | −0.45 | 0.27 | $9.13 \times 10^{-02}$ |
| HDL cholesterol, SD mg/dL | −0.08 | 0.21 | $7.10 \times 10^{-01}$ | −0.61 | 0.38 | $1.05 \times 10^{-01}$ | −0.14 | 0.24 | $5.75 \times 10^{-01}$ |
| Triglycerides, SD mg/dL | 0.06 | 0.21 | $7.89 \times 10^{-01}$ | 0.30 | 0.37 | $4.26 \times 10^{-01}$ | −0.16 | 0.24 | $5.09 \times 10^{-01}$ |
| Fasting glucose, mmol/L | 0.09 | 0.10 | $3.58 \times 10^{-01}$ | 0.15 | 0.18 | $4.23 \times 10^{-01}$ | 0.30 | 0.14 | **$3.64 \times 10^{-02}$** |
| Fasting insulin, log pmol/L | −0.01 | 0.11 | $9.58 \times 10^{-01}$ | −0.02 | 0.19 | $9.16 \times 10^{-01}$ | 0.18 | 0.14 | $1.96 \times 10^{-01}$ |
| HOMAB, log-HOMA | −0.06 | 0.11 | $5.96 \times 10^{-01}$ | 0.22 | 0.18 | $2.15 \times 10^{-01}$ | 0.14 | 0.15 | $3.61 \times 10^{-01}$ |
| HOMA-IR, log-HOMA | −0.09 | 0.13 | $4.82 \times 10^{-01}$ | 0.30 | 0.22 | $1.69 \times 10^{-01}$ | 0.37 | 0.18 | **$4.53 \times 10^{-02}$** |
| HbA1c, % | 0.05 | 0.11 | $6.69 \times 10^{-01}$ | −0.28 | 0.18 | $1.33 \times 10^{-01}$ | 0.04 | 0.16 | $8.02 \times 10^{-01}$ |
| Diastolic blood pressure, mmHg | 5.54 | 0.99 | **$2.37 \times 10^{-08}$** | −3.86 | 1.55 | **$1.26 \times 10^{-02}$** | −0.47 | 1.03 | $6.46 \times 10^{-01}$ |
| Systolic blood pressure, mmHg | 10.67 | 1.48 | **$4.72 \times 10^{-13}$** | 0.90 | 2.30 | $6.95 \times 10^{-01}$ | 3.61 | 1.54 | **$1.87 \times 10^{-02}$** |
| Coronary artery disease, log-odds | −0.10 | 0.32 | $7.47 \times 10^{-01}$ | 2.02 | 0.45 | **$9.26 \times 10^{-06}$** | 0.52 | 0.37 | $1.64 \times 10^{-01}$ |
| Type 2 diabetes, log-odds | 0.52 | 0.61 | $3.92 \times 10^{-01}$ | 1.52 | 1.06 | $1.51 \times 10^{-01}$ | 0.17 | 0.72 | $8.17 \times 10^{-01}$ |

Cluster 1: *CRHR1, SKOR2, KSR2, ASCL4, RERE*, and *ECE2*.
Cluster 2: *SHISA4, ADO, NRXN3, FNDC5*, and *GS1-259H13.13*.
Cluster 3: *HCRTR2, ALG10, ALG10B, PATJ, BTBD9, MTNR1B, AGAP1, RP11-6N13.1*, and *ZBTB5*.
Cluster-specific polygenic scores were calculated by summing the products of the daytime napping-increasing effect allele SNP multiplied by the scaled effect from the discovery GWAS using the GTX package in R.
Effect estimates (beta) are reported per additional daytime napping increasing effect allele.
Summary statistics for outcome traits were obtained from the Sleep Disorder Knowledge Portal (http://sleepdisordergenetics.org/) for sleep traits or publicly available data for cardiometabolic traits. Study characteristics for cardiometabolic traits are indicated in Supplementary Table 11.
Bolded P values indicate cardiometabolic traits with cluster-specific polygenic score associations $P < 0.05$.

narcolepsy. To test for such potential on-target cardiovascular side effects, we used missense variants in *HCRTR1* (A allele of rs2271933) and *HCRTR2* (A allele of rs2653349), both associated with more frequent daytime napping and daytime sleepiness[24], as proxies for pharmacologic inhibition of these proteins, and tested for associations with cardiovascular phenotypes in large GWAS (Supplementary Table 13). This analysis revealed no associations of the variants with cardiovascular outcomes, but showed opposing effects on systolic blood pressure at *HCRTR1* (−0.10 mmHg, 95% CI [−0.17, −0.04], $P = 1.00 \times 10^{-4}$) and *HCRTR2* (0.14 mmHg, 95% CI [0.07, 0.21], $P = 1.00 \times 10^{-04}$; Fig. 6). We further performed a hypothesis-free scan across 1402 ICD-code defined phenotypes in the UK Biobank[50] and found no variant-disease associations (Supplementary Fig. 5; Supplementary Data 13). The present human genetic evidence therefore does not support a net excess adverse cardiovascular risk from on-target inhibition of *HCRTR1* and *HCRTR2*, but suggests potential opposing effects on blood pressure regulation by the two receptors.

## Discussion

We comprehensively investigated the genomic influences of daytime napping using the largest discovery and replication sample sizes to date. We identified 123 independent loci in the UK Biobank with strong evidence of replication in 23andMe, an

independent study with different demographic characteristics. Variant effects were largely independent of BMI and sleep apnea, and the associations retained significance when GWAS was restricted to healthier participants, a strong determinant of 5-year mortality in the UK Biobank[51], suggesting that signals were not driven by poor health. In addition, despite higher prevalence of daytime napping among men compared to women[52], we identified only one sex-specific signal on the X chromosome, suggesting sex differences may be attributed to environmental factors or possibly rare genetic variants. Our results advance the understanding of the biology of daytime napping, refine the understanding of pleiotropy and causality in the relationship of napping with sleep and cardiometabolic traits, and inform pharmacologic investigations of orexin antagonism.

The identified variants highlight a central role for arousal-regulating neuropeptide signaling pathways in daytime napping propensity. Most prominent among these pathways was the well-established hypocretin arousal pathway[53] (including missense variants in *HCRTR1, HCRTR2,* and *TRPC6*). It is thus possible that orexin receptor agonism, a therapeutic strategy currently under investigation for narcolepsy, may have roles in the treatment of patients with more mild deficits in the arousal/wake drive system[54]. Additional pathways with known roles in sleep-wake biology in model organisms[55] include neuronal excitability driven by variation in the function of potassium channels and glutamate signaling, EGFR signaling pathway, and opioid signaling.

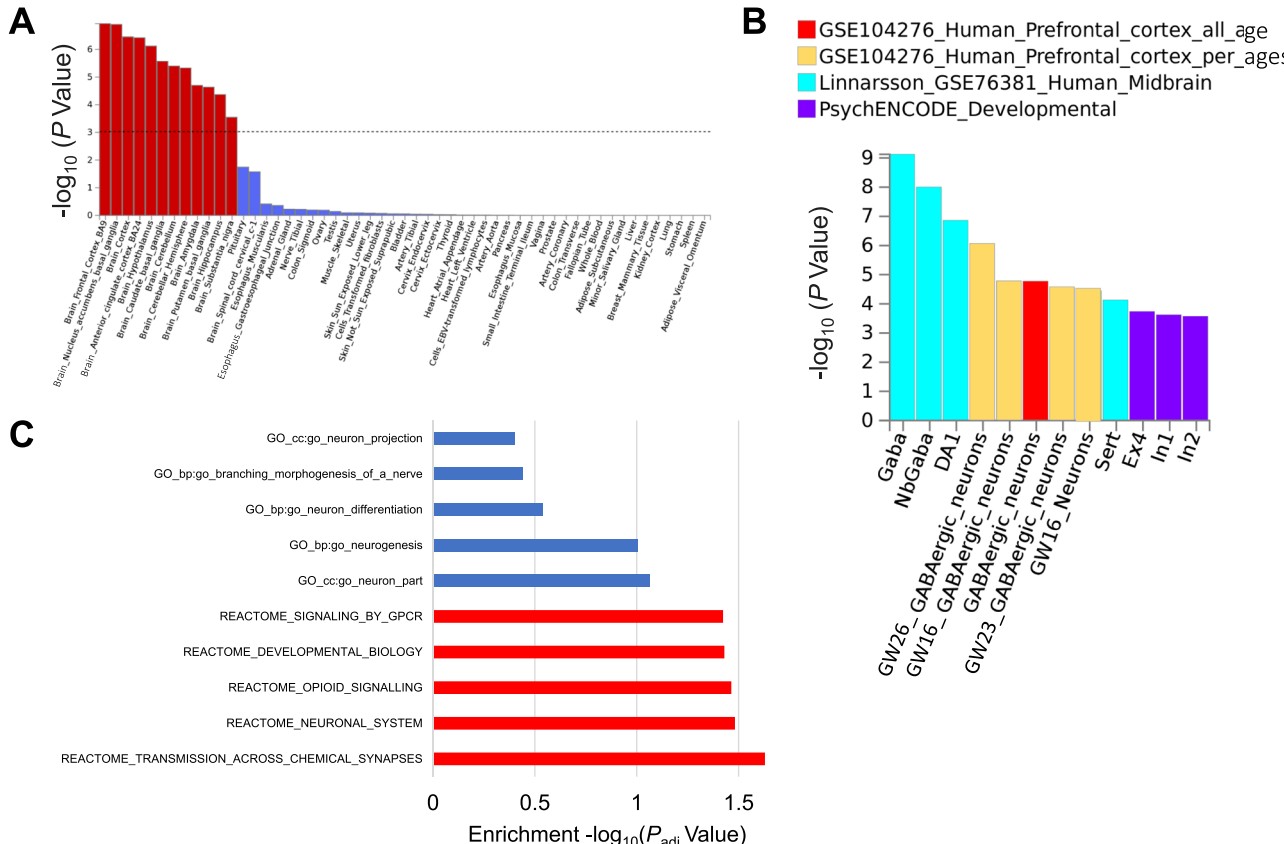

**Fig. 3 Tissue expression, single-cell, and pathway-based enrichment analyses for daytime napping. A** MAGMA tissue expression analysis using gene expression per tissue based on GTEx RNA-seq data for 53 specific tissue types. Significant tissues ($P < 9.43 \times 10^{-4}$) are shown in red. **B** Significant single-cell types from single-cell enrichment analyses using human brain datasets in FUMA. **C** Top pathways determined from analysis using MAGMA gene sets and Pascal (gene-set enrichment analysis using 1077 pathways from KEGG, REACTOME, BIOCARTA). Significant pathways are shown in red ($P_{adj} < 0.05$). All pathway and tissue expression analyses in this figure can be found in tabular form in Supplementary Table 10, Supplementary Data 8, 9.

Expression of genes under association peaks was most enriched in the frontal cortex, similar to observations for daytime inactivity duration[19], and other brain regions prominently implicated in sleep duration, timing, and quality traits[23,25,26]. Cross-trait clustering of the identified loci suggest at least three underlying physiologic mechanisms, including (1) propensity for longer sleep, (2) consequence of poor and disrupted sleep, and (3) napping concomitant with early sleep timing, potentially reflecting loss of function in arousal pathways. Notably, genetic links between daytime napping and sleep disorders, e.g., sleep apnea or restless legs syndrome, may be partially undetected by our study because of incomplete ascertainment of these disorders in the UK Biobank and the lack of available summary statistics in public repositories and databases. We found that the genetic architecture of daytime napping is shared with cardiometabolic diseases and traits, consistent with previous epidemiologic associations of more frequent daytime napping with increased cardiometabolic risk[7–13,56]. At the locus level, we observed colocalization of the daytime napping loci with daytime sleepiness, snoring, chronotype, and BMI loci at *PNOC* and *PATJ*, suggesting an obesity-hypersomnolence pathway[57]. Furthermore, colocalization of *FADS1* gene expression in the frontal cortex with the daytime napping signal suggests uncharacterized pleiotropic effects of lipid metabolism on sleep. Positive genome-wide genetic correlations were observed with multiple anthropometric, glycemic, and cardiometabolic traits, of which several correlations were attenuated after accounting for BMI. In a large health system-based clinical cohort, phenome-wide association analyses using a

daytime napping genome-wide polygenic score further supported associations with obesity and hypertension, in addition to other cardiometabolic diseases. Although daytime napping shares biological determinants with other sleep traits, most prominently daytime sleepiness[24], there were several genetic findings unique to daytime napping. There were 26/123 loci unique to daytime napping, with several other loci exhibiting stronger relationships with daytime napping relative to other traits (e.g., *KSR2* locus). The SNP-based heritability of daytime napping (11.9%) was almost double that previously reported for daytime sleepiness (6.9%)[24], and daytime napping variants were modestly attenuated in GWAS models accounting for daytime sleepiness. Although prior analyses related higher BMI to more frequent daytime sleepiness[24], we observed no such relationship with frequency of daytime napping. Taken together, these data suggest that daytime napping and daytime sleepiness should be considered related, but distinct features of the impaired arousal continuum.

A key clinical question is whether habitual daytime napping has causal effects on cardiometabolic health. Findings from our Mendelian randomization analyses suggest potentially deleterious effects of daytime napping frequency on cardiometabolic health, with effects on increased blood pressure and waist circumference. A causal effect of more frequent napping with higher blood pressure is consistent with earlier epidemiologic findings between self-reported and actigraphy-measured daytime napping and hypertension[58–60]. Mechanisms driving this relationship are unknown but may include detrimental effects of napping on nighttime sleep quality, or chronic effects related to transient

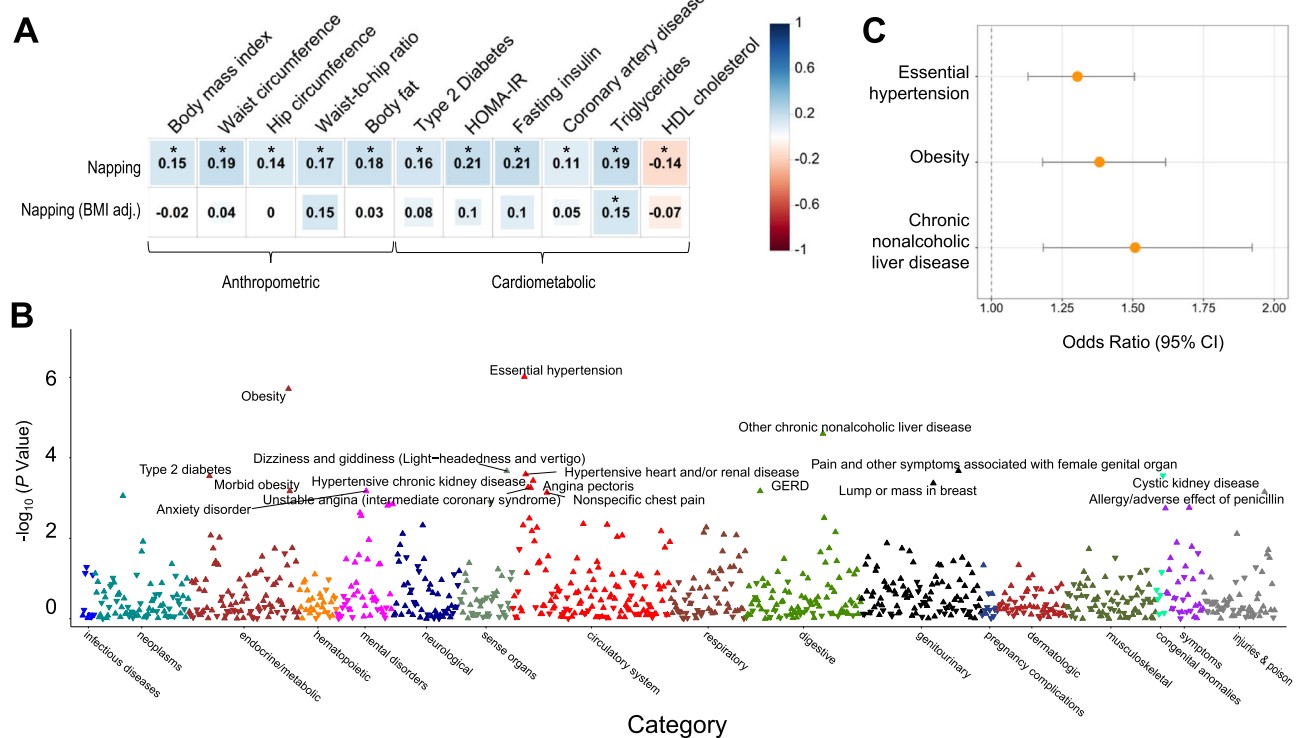

**Fig. 4 Genome-wide genetic architecture of daytime napping correlations and associations with diseases and traits. A** Shared genetic architecture between daytime napping and cardiometabolic diseases and traits. Linkage disequilibrium (LD) score regression estimates of genetic correlation ($r_g$) were obtained by comparing genome-wide association estimates for daytime napping (without and with BMI adjustment) with summary statistics estimates from 257 publicly available genome-wide association studies. Blue indicates positive genetic correlation and red indicates negative genetic correlation; $r_g$ values are displayed for significant correlations. Larger colored squares correspond to more significant $P$ values. Asterisk denotes significant false discovery rate (FDR) corrected $P$ values. Full genetic correlations for all 257 traits can be found in Supplementary Data 10. **B** Manhattan plot of phenome-wide association findings for daytime napping genome-wide polygenic score in Mass General Brigham Biobank ($n = 23,561$). The x-axis is color-coded phecodes organized by broad disease categories and the y-axis is $P$ value of association ($-\log_{10} P$). The horizontal red line depicts phenome-wide significance using Bonferroni correction for all tested diseases (951 diseases), and the horizontal blue line depicts phenome-wide significance using FDR correction. Upward arrows denote positive associations (OR > 1), and downward arrows denote inverse associations (OR < 1). Full results for all 951 diseases can be found in Supplementary Data 11. **C** Cross-sectional association between quartile 10 and quartile 1 (reference group) of daytime napping genome-wide polygenic score and essential hypertension, obesity, and chronic nonalcoholic liver disease in the Mass General Brigham Biobank ($n = 23,561$). Error bars represent the 95% confidence intervals for association.

evening blood pressure surges following daytime napping[61,62]. Similarly, mechanisms underlying the link between daytime napping and body fat distribution are poorly understood[63]. Although results from the MR Egger sensitivity analysis of waist circumference on daytime napping were inconsistent with findings from our primary MR analysis, the genetic overlap we demonstrated with BMI indicates that the Egger analysis may be biased by violation of the "instrument strength independent of direct effect (INSIDE)" assumption[64]. Polygenic scores of each napping subtype showed heterogeneous associations with cardiometabolic outcomes across clusters, including associations with higher blood pressure for cluster 1, and other adiposity traits for clusters 2 and 3. Exploring causal relationships with biologically distinct subtypes of daytime napping will be important to understand the beneficial or detrimental role of different aspects of napping biology with disease outcomes.

We leveraged coding variation in *HCRTR1* and *HCRTR2* to predict the cardiovascular consequences of long-term pharmacologic modulation of orexin receptors. We found no net effect of these genetic proxies on cardiovascular outcomes, nor on any ICD-code defined disease outcomes in a PheWAS. These results predict that pharmacologic agonism or antagonism of orexin receptors therapies is unlikely to increase the risk of cardiovascular disease. An association of *HCRTR1* and *HCRTR2* with blood pressure was observed, however, the direction of effect differed for the two variants. This suggests a neutral net blood pressure effect of dual orexin receptor antagonism, and more broadly suggests pleiotropic effects of these proteins on blood pressure regulation. However, it is possible that these genetic variants do not proxy peripheral effects of HCRTR1 and HCRTR2 inhibition (e.g., bone marrow)[49]. The application of PheWAS to study on-target side effects of sleep medications sets the stage for future use of these genetic proxies to understand the health consequences of orexin receptor modulation.

Our analyses are limited by the crude assessment of daytime napping frequency via questionnaire with no information on duration or timing. Our effort to partly validate the specificity of our discovered loci from self-report with an objectively determined daytime napping behavior from accelerometer was likely limited as a result of phenotypic differences between self-report and accelerometer (self-report was based on daytime napping frequency whereas accelerometer measures were based on daytime inactivity duration in the absence of sleep diaries; Pearson correlation $r^2 = 0.17$), relatively smaller sample size in the accelerometer subsample ($n = 85,670$), or lapsed time between measurements as the accelerometer was worn between 2.8 and

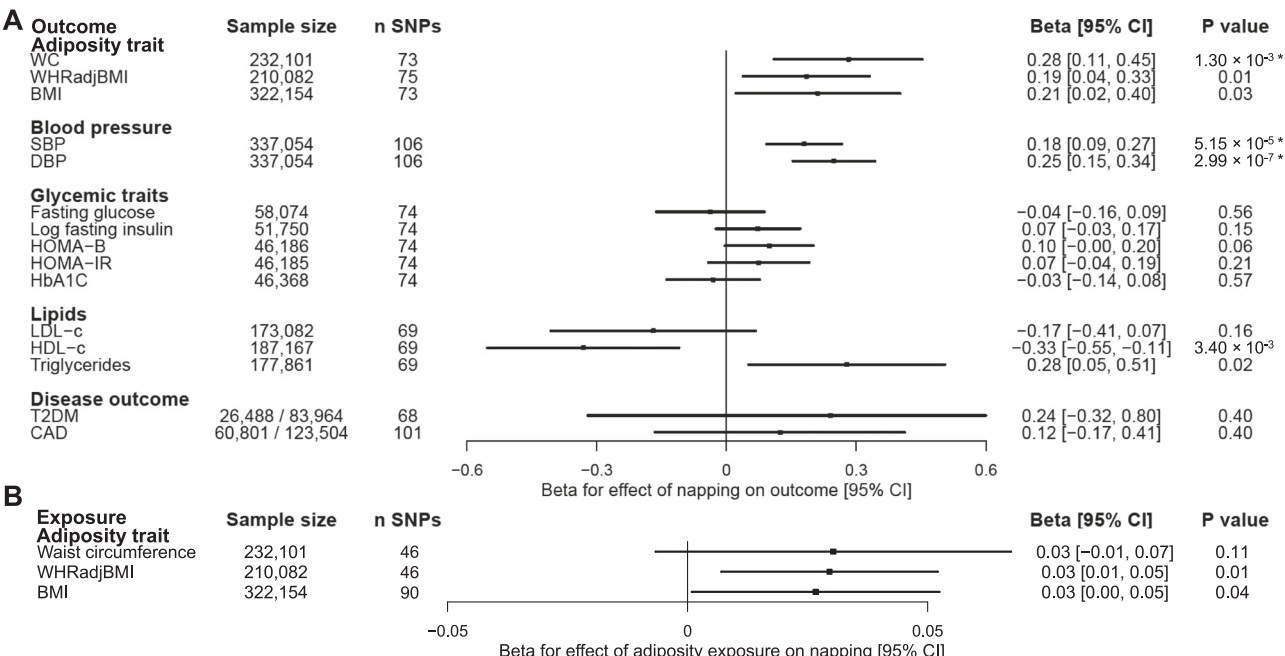

**Fig. 5 Mendelian randomization supports a causal effect of daytime napping on higher blood pressure and waist circumference.** The MR estimates were calculated using the random-effects inverse-variance weighted method and represent the effect of a one-unit increase in napping category (never, sometimes, usually). Sample sizes reflect either the total sample size (for continuous outcomes) or number of cases and controls (for binary outcomes). **A** IVW effect estimates for more frequent daytime napping on cardiometabolic outcomes and risk factors. A unit increase in the adiposity and blood pressure measurement represents a standard deviation increase in the corresponding trait. Black boxes show effect estimates, and surrounding lines display 95% confidence intervals. All P values are two-sided. **B** IVW effect estimates for the effect of adiposity traits on daytime napping frequency. Black boxes show effect estimates, and surrounding lines display 95% confidence intervals. All P values are two-sided. * significant at Bonferroni-corrected alpha threshold and robust in sensitivity analyses. BMI body-mass index, CAD coronary artery disease, CI confidence interval, DBP diastolic blood pressure, HOMA homeostatic model assessment of insulin resistance, HOMAB homeostasis model assessment of β-cell function, LDL low-density lipoprotein, HDL high-density lipoprotein, OR odds ratio, SBP systolic blood pressure, SNP single-nucleotide polymorphism, T2DM type 2 diabetes mellitus, WC waist circumference, WHRadjBMI waist-to-hip ratio adjusted for BMI.

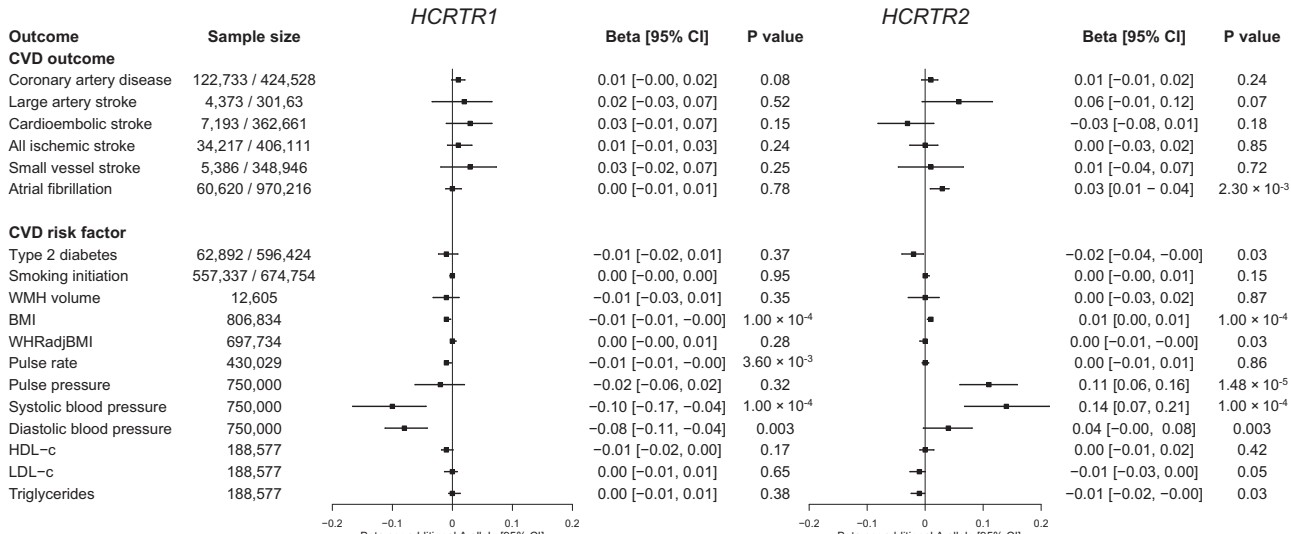

**Fig. 6 Cardiovascular risk factor and disease associations of missense variants in *HCRTR1* (rs2271933) and *HCRTR2* (rs2653349), which encode targets of Suvorexant, an FDA-approved sleep medication with an unknown cardiovascular safety profile.** Sample size either reflects the total number of subjects (for continuous traits), or the number of cases and controls (for binary traits) that were included in each of the genome-wide association studies. All associations are oriented to the napping-increasing allele of the variants. Additional details regarding the included studies are provided in Supplementary Table 13 and Supplementary Data 13. Black boxes show Mendelian randaomization effect estimates and surrounding lines display 95% confidence intervals. BMI body-mass index, CI confidence interval, CVD cardiovascular disease, HDL high-density lipoprotein cholesterol, LDL low-density lipoprotein cholesterol, OR odds ratio, WMH white matter hyperintensities, WHR waist-to-hip ratio.

9.7 years after study baseline. Replication of most loci and the specific association with daytime activity duration, but not other accelerometer measures, however, support our findings. The low participation rate of the UK Biobank at 5.5% may have introduced selection bias. However, consistency of the genetic signals between the UK Biobank and 23andMe, an independent study with different demographic, and various findings with the Mass General Brigham Biobank, an independent clinical cohort, supports the generalizability of our findings. In addition, the identification of variants in pathways with known relevance to sleep (e.g., *HCRTR1* and *HCRTR2*) suggests that the GWAS is capturing true biological signal. Nonetheless, continued evaluation in other demographics, including age-groups and ancestries, is necessary. It remains possible that rare and structural variation have an important contribution to the genetic architecture of daytime napping, however, these data were not tested in the present analysis. In addition, our analysis was limited in scope to cardiometabolic health, and future studies should evaluate the impact of daytime napping on other health outcomes including mental health. Finally, despite consistency in Mendelian randomization estimates, these analyses require strong, unverifiable assumptions for the determination of causality and therefore require confirmation in randomized controlled trials of sleep interventions. Further dissection of the heterogeneity of daytime napping is necessary to determine which types of daytime napping behavior are most detrimental to cardiometabolic health. In addition, future analyses investigating sex heterogeneity in daytime napping frequency is warranted. In summary, our genetic analyses contribute important insight into the biology and cardiometabolic consequence of habitual daytime napping in adults.

## Methods

**UK Biobank**. The UK Biobank is a large population-based study established to facilitate detailed investigations of the genetic and lifestyle determinants of a wide range of phenotypes[65]. Data from >500,000 participants living in the United Kingdom who were aged 40–69 and living <25 miles from a study center participated in the study between 2006 and 2010. Extensive phenotypic data were self-reported upon baseline assessment by participants using touchscreen tests and questionnaires and at nurse-led interviews. The UK Biobank study was approved by the National Health Service National Research Ethics Service (ref. [11]/NW/0382), and all participants provided written informed consent to participate. The current study was conducted under UK Biobank application 6818.

**Daytime napping, covariates, and other self-reported and objectively measured sleep traits**. At baseline assessment, all study participants reported their daytime napping frequency (*n* = 501,646). Participants were asked Do you have a nap during the day?, with responses Never/rarely, Sometimes, Usually, Prefer not to answer. Responses were treated as a continuous variable in the GWAS. *Prefer not to answer* responses were set to missing. Participants further self-reported age, gender, sleep duration, chronotype, insomnia symptoms, sleep apnea, smoking, and overall health. Weight, height, and waist circumference were measured and body-mass index (BMI) was calculated as weight (kg)/height$^2$ (m$^2$). Systolic and diastolic blood pressure were measured at baseline and the average of two automated readings was used. Socio-economic status was represented by the Townsend deprivation index based on national census data immediately preceding participation in the UK Biobank. Assessment season was determined from baseline assessment visit date and categorized as 1 for winter [January–March], 2 for spring [April–June], 3 for summer [July–September], and 4 for fall [October–December], as previously conducted[66]. Participants rated their overall health in response to the question, In general how would you rate your overall health?, with responses excellent, good, fair, poor, do not know, and prefer not to answer. Cases of sleep apnea were determined from self-report during nurse-led interviews or health records using International Classification of Diseases (ICD)-10 codes for sleep apnea (G47.3). For each participant, a modified STOP-BANG risk scale[21] we have previously developed for sleep apnea in the UK Biobank to account for undiagnosed sleep apnea, was calculated[67]. The modified STOP-BANG risk scale for sleep apnea is missing the question, Has anyone observed you stop breathing during sleep? and replacing neck circumference with waist circumference dichotomized to the threshold for metabolic syndrome. Insomnia symptoms were ascertained from self-report to the question, Do you have trouble falling asleep at night or do you wake up in the middle of the night? with responses never/rarely, sometimes, usually, prefer not to answer. Participants who responded usually were set as insomnia cases, and remaining participants were set as controls. Smoking status

(never, former, current) was further self-reported. Missing covariates were imputed by using sex-specific median values for continuous variables (i.e., BMI and Townsend index).

A subset of 103,711 participants from the UK Biobank wore actigraphy devices (Axivity AX3) for up to 7 days, ~2.8–9.7 years after their study baseline visits. Details on quality control and data processing have been described previously[19,68]. The following sleep measures were derived by processing raw accelerometer data: daytime inactivity duration, sleep duration, sleep efficiency, number of sleep bouts within the sleep period time window, sleep midpoint, midpoint of the least-active 5 h of the day (L5 timing), and midpoint of the most-active 10 h of the day (M10 timing). Specifically, daytime inactivity duration was estimated by the total daily duration of estimated bouts of inactivity that fell outside of the sleep period time window. These inactivity bouts are any inactivity lasting ≥30 min. Inactivity bouts that are <60 min apart are combined to form inactivity blocks. This measure captures very inactive states such as napping and wakeful rest but not inactivity such as sitting and reading or watching television, which are associated with a low but detectable level of movement[19].

**Genome-wide association study for daytime napping in UK Biobank**. Genotyping was performed by the UK Biobank, and genotyping, quality control, and imputation procedures are described in detail previously[69]. In brief, blood, saliva, and urine were collected from participants, and DNA was extracted from the buffy coat samples. Participant DNA was genotyped on two arrays, UK BiLEVE and UK Biobank Axiom with >95% common content and genotypes for ~800,000 autosomal SNPs were imputed to two reference panels. Genotypes were called using Affymetrix Power Tools software. Sample and SNPs for quality control were selected from a set of 489,212 samples across 812,428 unique markers. Sample quality control (QC) was conducted using 605,876 high-quality autosomal markers. Samples were removed for high missingness or heterozygosity (968 samples) and sex chromosome abnormalities (652 samples). Genotypes for 488,377 samples passed sample QC (~99.9% of total samples). Marker-based QC measures were tested in the European ancestry subset (*n* = 463,844), which was selected based on principal components of ancestry. SNPs were tested for batch effects (197 SNPs/batch), plate effects (284 SNPs/batch), Hardy–Weinberg equilibrium (572 SNPs/batch), sex effects (45 SNPs/batch), array effects (5417 SNPs), and discordance across control replicates (622 on UK BiLEVE Axiom array and 632 UK Biobank Axiom array; *P* value <10$^{-12}$ or <95% for all tests). For each batch (106 batches total) markers that failed at least one test were set to missing. Before imputation, 805,426 SNPs pass QC in at least one batch (>99% of the array content).

Population structure was captured by principal component analysis on the samples using a subset of high-quality (missingness < 1.5%), high-frequency SNPs (>2.5%) (~100,000 SNPs) and identified the subsample of white British descent. In addition to the calculated population structure by the UK Biobank, we locally further clustered subjects into four ancestry clusters using K-means clustering on the principal components, identifying 453,964 subjects of European ancestry. For the current analysis, individuals of non-white ethnicity were excluded to limit confounding effects. The UK Biobank centrally imputed autosomal SNPs to UK10K haplotype, 1000 Genomes Phase 3, and Haplotype Reference Consortium (HRC). Autosomal SNPs were pre-phased using SHAPEIT3 and imputed using IMPUTE4. In total ~96 million SNPs were imputed. Related individuals were identified by estimating kinship coefficients for all pairs of samples, using only markers weakly informative of ancestral background.

Genetic association analysis for daytime napping (never/rarely, sometimes, and always) was performed in related subjects of European ancestry with self-reported daytime napping data (*n* = 452,633) using BOLT-LMM[70] linear mixed models and an additive genetic model adjusted for age, sex, 10 principal components of ancestry, genotyping array and genetic correlation matrix with a maximum per SNP missingness of 10% and per sample missingness of 40%. We used a SNP imputation quality threshold of 0.80 and a MAF threshold of 0.001. X chromosome data were imputed and analyzed separately using the same analytical approach in BOLT-LMM as was done for analysis of autosomes. A rare signal at *IGSF1* on the X chromosome driven by one rare variant (rs189568547; MAF = 0.006) was identified, potentially attributed to genotyping artifact or false-positive association and therefore was excluded.

Trait heritability was calculated as the proportion of trait variance due to additive genetic factors measured in this study using BOLT-REML[70], to leverage the power of raw genotype data together with low-frequency variants (MAF ≥ 0.001). Lambda inflation (λ) values were calculated using GenABEL in R, and estimated values were consistent with those estimated for other highly polygenic complex traits. Furthermore, follow-up GWAS for daytime napping were conducted using BOLT-LMM[70] and included sensitivity analyses restricted to participants self-reporting excellent or good overall health[51] (*n* = 338,764), GWAS adjusting for BMI in addition to baseline adjustments, GWAS adjusting for BMI and BMI × BMI in addition to baseline adjustments, to account for sleep apnea, GWAS excluding participants with diagnosed sleep apnea (*n* = 5553 excluded) and GWAS adjusting for a modified STOP-BANG risk scale[21,67] in addition to baseline adjustments, GWAS adjusting for self-reported daytime sleepiness, and sex-stratified GWAS (male *n* = 207,108; female *n* = 245,525).

Distinct genomic risk loci were defined using FUMA v1.3.3 on the basis of genome-wide significance (*P* < 5 × 10$^{-8}$) and pairwise independence (*r*$^2$ < 0.6)

within a 1 Mb window. Annotation of the lead variants, including predicted sequence consequence, was obtained from the FUMA output. We determined the PICS probability for each lead variant being the causal variant at the locus[71].

For the 123 lead variants, we tested for gene-by-season interaction in PLINK[72] among unrelated participants of white British ancestry ($n = 337,409$) using linear regression and an additive genetic model. Interaction analyses were adjusted for age, sex, 10 principal components of ancestry, genotyping array, and season to determine SNP interaction with season on daytime napping. In addition, for each lead variant, corresponding summary statistics for other self-reported and accelerometer-derived sleep measures were obtained from the Sleep Disorder Knowledge Portal (http://sleepdisordergenetics.org/). As earlier UK Biobank GWASs were restricted to HRC-imputed variants, if the lead signal was unavailable, a proxy SNP ($r^2 > 0.8$) was used instead.

**23andMe, Inc. replication.** 23andMe, Inc. is a personal genetics company. DNA extraction and genotyping were performed on saliva samples by National Genetics Institute, a CLIA licensed clinical laboratory and a subsidiary of Laboratory Corporation of America. Samples were genotyped on one of five genotyping platforms. Samples that failed to reach 98.5% call rate were re-analyzed. A single unified imputation reference panel was created by combining the May 2015 release of the 1000 Genomes Phase 3 haplotypes[73] with the UK10K imputation reference panel[74]. For each chromosome, Minimac3[75] was used to impute the reference panels against each other, reporting the best-guess genotype at each site. Ancestry was determined through an analysis of local ancestry[76]. A principal component analysis was performed independently for each ancestry, using ~65,000 high-quality genotyped variants present in all five genotyping platforms. In addition, a maximal set of unrelated individuals was chosen for each analysis using a segmental identity-by-descent estimation algorithm. All individuals included in the analyses provided informed consent and answered surveys online according to human subject protocol, which was reviewed and approved by Ethical & Independent Review Services, a private institutional review board (http://www.eandireview.com).

For the present daytime napping replication, we restricted analyses to 541,333 participants of European ancestry with survey responses to a question on frequency of daytime napping. Participants were asked, How many days per week do you take naps during the day? (15 min or more) with a response on a continuous scale. Responses in days per week were scaled to never/rarely if 0 or 1 ($n = 267,271$), sometimes if 2 to 5 ($n = 232,868$), and usually if 6 or 7 ($n = 41,194$) to more closely resemble the UK Biobank categories. Replication for the 123 daytime napping loci or proxy for lead SNP ($r^2 > 0.80$) were generated through linear regression (using an additive model) of the phenotype against the genotype, adjusting for age, sex, the first four principal components, and a categorical variable representing genotyping platform. Furthermore, meta-analysis of UK Biobank and 23andMe associations for the daytime napping loci was performed using METAL[77] by weighting effect-size estimates using the inverse of the corresponding squared standard errors (version released 25 March 2011).

**Colocalization.** To identify genomic regions which harbor causal variants that influence multiple sleep traits, we performed multi-trait colocalization using the Hypothesis Prioritization Colocalization (HyPrColoc) package[28]. This package performs multi-trait colocalization using a computationally efficient algorithm that facilitates colocalization of large numbers of traits. To identify clusters of colocalized traits, we implemented the branch and bound divisive clustering algorithm using GWAS summary statistics for the following sleep traits in the UK Biobank: sleep duration ($n = 446,118$)[26], insomnia symptoms ($n = 129,270$ cases/108,357 controls)[25], chronotype ($n = 449,734$)[23], snoring ($n = 421,466$), ease of awakening ($n = 451,872$), and daytime sleepiness ($n = 452,071$). Although these GWAS were conducted in the UK Biobank, the algorithm is robust to inclusion of studies with overlapping participants[28]. Colocalization analysis was performed in pre-defined, approximately independent LD blocks across the genome (1.6 Mb on average)[78]. We used the default variant-level prior probability of a SNP associated with a trait of $p_1 = 1 \times 10^{-4}$ (prior probability of a SNP being associated with one trait) and $y = 0.98$ (1 − prior probability of a SNP being associated with an additional trait given that the SNP is associated with at least 1 other trait). With these settings, 1 in 200,000 variants are expected to be causal for two traits. Consistent with prior work[28], we conservatively set both the regional and alignment probabilities to 0.80 so that a cluster of colocalized traits would only be identified if $P_R P_A > 0.64$. The outputs from the algorithm include: (i) colocalized traits, (ii) the posterior probability of colocalization, (iii) the regional association probability (a measure of degree of shared association, analogous to a phenome-wide association study), (iv) the candidate causal variant, and (v) the proportion of the posterior probability of colocalization explained by the genetic variant (interpreted as a multi-trait fine-mapping probability). We report loci with posterior probability (pp) for colocalization >0.7, as this cutoff corresponds to a false discovery rate of <5%[28].

We performed two additional colocalization analyses. Using summary statistics from a meta-analysis ($n \sim 700,000$) of UK Biobank and the GIANT consortium[79], we performed genome-wide colocalization of naps with BMI. To link gene expression to the naps associations, we performed colocalization for all genes located within 1 MB of the top signals identified in the naps GWAS. We used summary statistics for expression quantitative trait loci (eQTL) associations identified in the Genotype-Tissue Expression project v7[80]. We prioritized gene

expression in the frontal cortex, which was identified by FUMA analysis of GTEX v7 to be the most highly enriched tissue for the naps signals.

**Conditional analysis.** To determine independence of the daytime sleepiness and napping association signals at the *HCRTR2* locus, we applied the GCTA COJO algorithm to perform conditional analyses[81]. We used the UK Biobank sample as the LD reference panel and considered a 1 Mb window surrounding the lead *HCRTR2* SNP in the napping GWAS (position 6:55142337). We conditioned on rs2653349 using the --cojo-cond function.

**Bayesian nonnegative matrix factorization (bNMF) clustering and association.** We applied the bNMF clustering algorithm[42,82,83] with the aim of collapsing identified naps loci into subgroups of variants based on patterns of association with other sleep traits. The inputs for the bNMF algorithm were the set of the 122 naps GWAS signals (rs10639111 was not included due to missing proxy SNP in association analyses for other sleep traits) oriented to naps-increasing alleles and corresponding association statistics for 17 self-reported and accelerometer-derived sleep traits from the UK Biobank. We generated standardized effect sizes for variant-trait associations from GWAS by dividing the estimated regression coefficient by the standard error, using the UK Biobank summary statistic results (variant-trait association matrix [122 by 17]). To enable an inference for latent overlapping modules or clusters embedded in variant-trait associations, we modified the existing bNMF algorithm to explicitly account for both positive and negative associations as was done previously[42,83].

The defining features of each cluster were determined by the most highly associated traits, which is a natural output of the bNMF approach. bNMF algorithm was performed in R for 1000 iterations with different initial conditions, and the maximum posterior solution at the most probable number of clusters was selected for downstream analysis (i.e., $k = 3$ for 63% of 1000 iterations in this analysis, with those same 3 clusters present in an additional 34% of iterations with $k = 4$). The results of the bNMF algorithm provide cluster-specific weights for each variant and trait. Variants and traits defining each cluster were based on a cutoff of weighting of 1.09, which was determined by the optimal threshold to define the beginning of the long-tail of the distribution of cluster's weights across all clusters (top 5% were considered to be significant).

We compared our clusters from the bNMF algorithm using hierarchical cluster analysis, as was previously conducted for daytime sleepiness[24]. Briefly, the analysis uses the pairwise Euclidean distance between the 122 loci z-scores with the 17 self-reported and accelerometer-derived sleep traits.

**Functional annotations of SNPs and pathway and tissue-enrichment analyses.** Functional annotation was carried out using ANNOVAR in FUMA[84]. Missense variants of interest were further mapped to protein domains using UniProt[85]. Pathway analysis was conducted using MAGMA[45] gene-set analysis in FUMA[44], which uses the full distribution of SNP $P$ values and is performed for curated gene sets and GO terms obtained from MsigDB (total of 15,481 pathways). A significance threshold was set after Bonferroni correction accounting for all pathways tested ($P < 0.05/15,481$). Gene-based analysis was also performed using Pascal[43]. Pascal gene-set enrichment analysis uses 1077 pathways from KEGG, REACTOME, BIOCARTA databases, and a significance threshold was set after Bonferroni correction accounting for 1077 pathways tested ($P < 0.05/1077$). We performed single-cell enrichment analysis (Cell Type function) in FUMA[44] using our MAGMA gene analysis result and multiple human-specific single-cell expression datasets[86].

**Genetic correlations with publicly available traits and other sleep traits.** Genome-wide genetic correlation analysis were calculated using the implementation of cross-trait LD Score Regression (LDSC)[87–89] in LDHub[88]. This was conducted using all SNPs from the UK Biobank discovery GWAS found in HapMap3 and included publicly available data from 257 published genome-wide association studies. LDSC estimates genetic correlation between two traits from summary statistics (ranging from −1 to 1) using the fact that the GWAS effect-size estimate for each SNP incorporates effects of all SNPs in LD with that SNP, SNPs with high LD have higher statistics than SNPs with low LD, and a similar relationship is observed when single study test statistics are replaced with the product of z-scores from two studies of traits with some correlation. Significance was considered at the Bonferroni correction for all tests performed ($P < 0.05/257$ tests). In addition to publicly available summary statistics from LDHub, we also used publicly available summary statistics from earlier UK Biobank GWASs for self-reported and accelerometer-derived sleep traits from the Sleep Disorder Knowledge Portal (http://sleepdisordergenetics.org/) and computed genome-wide genetic correlations using LDSC[87–89]. Finally, we calculated genetic correlations between the sex-specific napping GWAS to determine the similarity in male and female genetic architecture.

**Phenome-wide association study in the Mass General Brigham Biobank.** The Mass General Brigham Biobank (formerly Partners Biobank) is a hospital-based cohort study from the Mass General Brigham healthcare network in Boston, MA with electronic health record (EHR) and genetic data. Recruitment for the Mass General Brigham Biobank launched in 2010 and is active at participating clinics at Brigham and Women's Hospital, Massachusetts General Hospital, Spaulding

Rehabilitation Hospital, Faulkner Hospital, McLean Hospital, Newton-Wellesley Hospital, and North Short Medical Center. All patients provided consent upon enrollment and the study protocol was approved by Mass General Brigham Institutional Review Board. To date (07/2019), a total of 104,965 subjects were consented.

Genomic data for 30,683 participants were generated with the Illumina Multi-Ethnic Genotyping Array. The genotyping data were harmonized, and quality controlled with a three-step protocol, including two stages of genetic variant removal and an intermediate stage of sample exclusion. The exclusion criteria for variants were: (1) missing call rate ≥0.05, (2) MAF < 0.001, and (3) deviation from Hardy–Weinberg equilibrium ($P < 10^{-6}$). The exclusion criteria for samples were: (1) sex discordances between the reported and genetically predicted sex, (2) missing call rates per sample ≥0.02, (3) subject relatedness (pairs with estimated identity-by-descent ≥0.125, from which we removed the individual with the highest proportion of missingness), and (4) population structure showing more than four standard deviations within the distribution of the study population, according to the first four principal components. Phasing was performed with SHAPEIT2[90] and then imputations were performed with the Haplotype Reference Consortium Panel[91] using the Michigan Imputation Server[75]. Written consent was provided by all study participants. Approval for analysis of Biobank data was obtained by Mass General Brigham IRB, protocol #2018P002276.

Participant ancestry was determined using TRACE[92] and the Human Genome Diversity Project (HGDP)[93] as a reference panel. Principal component analysis outliers were determined by using a principal component analysis projection of the study samples onto the HGDP reference samples, and subsequently excluded from analysis. To correct for population stratification, we computed principal components using TRACE[92] in the subset with genetically European ancestry. Furthermore, sample relatedness was determined using PLINK[72], and subsequently one sample from each related pair was excluded.

In aggregate, participants had a total of 7,422,726 ICD-9 and ICD-10 diagnostic codes corresponding to 784,878 instances of phecodes with at least 2 distinct diagnostic codes. The most prevalent codes were 401.1 (essential hypertension: $n = 11,397$ cases) and 745 (pain in joint: $n = 10,333$ cases). A total of 951 distinct phecodes had at least 100 cases in the biobank.

We generated a genome-wide polygenic score (GPS) for each individual by summing naps-increasing risk alleles across the genome, each weighted by the beta estimate for that allele from the discovery GWAS, using PRSice[94]. Of the 13,304,132 SNPs, 18,310 duplicated variants and 1,856,569 ambiguous variants were excluded, and a total of 11,429,253 SNPs remained. At each site, clumped SNPs based on association $P$ value (the variant with the smallest $P$ value within a 250 kb range) were retained and all those in linkage disequilibrium, $r^2 > 0.1$, were removed. Following LD clumping, the GPS included 995,188 SNPs.

A total of 20,054,591 physician diagnoses were obtained for genotyped participants ($n = 30,683$) as determined from EHR. Same-day duplicated diagnoses ($n = 8,265,731$), non-ICD-9/10 codes ($n = 466,866$), codes from participants of non-European ancestry ($n = 2,968,741$) were removed, and a total of 8,353,253 ICD-9/10 diagnoses were kept in the analysis. Similar ICD-9 and ICD-10 were consolidated and then further collapsed to 1857 phecodes based on clinical similarity[95]. A total of 88.9% of the 8,353,253 ICD-9/10 codes mapped to a phecode. Participants with at least 2 codes for a specific phecode were considered cases for that respective category, whereas participants with no relevant code for that category were considered controls. Codes with at least 100 prevalent cases were kept in the analysis. The association between the daytime napping GPS and each of 951 disease code was tested using logistic regression with adjustments for age, sex, genotyping array, and 5 principal components, using the PheWAS R package[96]. Phenome-wide significance was considered at the Bonferroni threshold for 951 tested diseases outcomes and a less stringent FDR correction.

**Daytime napping and cluster-specific polygenic score associations with cardiometabolic and sleep traits**. We tested associations between daytime napping polygenic scores comprised of all variants (123 loci) and sub-scores restricted to cluster-specific variants (3 clusters) with a range of cardiometabolic traits using publicly available data (listed in Supplementary Table 11) and other sleep traits (for cluster-specific polygenic scores only) using data from the Sleep Disorder Knowledge Portal. We generated weighted polygenic scores calculated by summing the products of the daytime napping-increasing allele SNP multiplied by the scaled effect from the discovery GWAS using the GTX package in R[97]. Results are effect estimates per additional effect allele for more daytime napping.

**Mendelian randomization**. Mendelian randomization (MR) can be conceptualized as a naturally randomized experiment whereby individuals are randomized to more or less liability for an exposure on the basis of their inherited genetic variation. This approach rests on the random assortment of alleles at gametogenesis, which substantially reduces the effect of confounding on causal estimates, and eliminates the potential for reverse causal effects of outcomes on the exposure. We created a genetic instrument from the lead daytime napping variants. These variants were further clumped at a between-SNP $r^2 < 0.01$. To facilitate analyses, we utilized the TwoSampleMR package[98] to extract and harmonize data from outcome GWAS on a range of cardiometabolic traits of interest. For all cardiometabolic traits, we utilized two-sample MR, where the outcome GWAS did not overlap with the naps GWAS. When variants were not in the outcome dataset, we identified variants in

linkage disequilibrium with the top variant at $r^2 > 0.80$ using the 1000 G European reference data integrated into MRBase. Datasets were harmonized to match effect and reference alleles, and we attempted to match strand ambiguous alleles by allele identity and frequency when possible (MAF < 0.42). An analogous approach was taken for reverse MR of adiposity measures (waist circumference, WHRadjBMI, and BMI) on daytime napping.

In the case of systolic blood pressure (SBP) and diastolic blood pressure (DBP), for which independent, non-overlapping summary statistics are not readily available, we undertook a split-sample MR approach[99] whereby we randomly split the UK Biobank sample of unrelated participants of White British ancestry into two subsets. We then re-estimated genetic associations of napping with the top variants identified in the discovery GWAS within each subset, as well as the association of those variants with SBP and DBP within each subset. In order to reduce regression dilution bias, SBP and DBP were averaged over two measurements and adjusted for self-reported antihypertensive use as done in prior GWAS[100]. We then performed MR utilizing exposure and outcome associations measured in different strata (e.g., napping associations in stratum 1 on diastolic blood pressure in stratum 2). MR effect estimates of daytime napping on blood pressure were combined across the two estimates using fixed-effects meta-analysis, and standardized using the sample standard deviations for SBP (11.25 mmHg) and DBP (20.65 mmHg) in the UK Biobank. As a sensitivity analysis, we used daytime napping variant association statistics from the 23andMe replication sample as the exposure, and a meta-analysis including UK Biobank and the International Consortium for Blood Pressure ($n \sim 750,000$) as the outcome[48].

After data harmonization, we used the random-effects inverse-variance weighted (IVW) method as the main analytic approach. To account for multiple comparisons, we used a conservative Bonferroni-adjusted alpha threshold (0.05/19 = 0.0026). As the IVW approach assumes no unbalanced horizontal pleiotropy, we utilized a range of sensitivity analyses robust to violations of this assumption: MR Egger[101], the simple and weighted median[102], MR-PRESSO[103], and multivariable Mendelian randomization. MR Egger models a pleiotropy parameter by fitting an intercept term and adjusts the causal estimates accordingly. Estimation of this additional parameter greatly reduces power in the Egger regression. The median estimators yield valid causal effects provided that <50% of the information comes from invalid instrumental variables. Regression-based multivariable MR analyses were performed to adjust the napping proxies for their associations with insomnia and sleep duration[25,26,104]. We considered consistent effects across multiple methods to strengthen causal evidence.

**Phenome-wide association study of HCRTR1 and HCRTR2**. To assess whether missense variants in *HCRTR1* and *HCRTR2* (rs2271933 and rs2653349) associated with cardiovascular outcomes and risk factors, we extracted variant associations from the largest available GWAS for these phenotypes (Supplementary Table 13). As a broader investigation, we used data from a phenome-wide association study of 1402 ICD-code based phenotypes in UK Biobank[50], accessed through the following web browser: http://pheweb.sph.umich.edu/SAIGE-UKB/ (Supplementary Data 13).

**Reporting summary**. Further information on research design is available in the Nature Research Reporting Summary linked to this article.

## Data availability
Summary GWAS statistics are publicly available at The Sleep Disorder Knowledge Portal webpage: http://sleepdisordergenetics.org/.

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

## Acknowledgements

This research has been conducted using the UK Biobank Resource (application 6818). We would like to thank the participants and researchers from the UK Biobank who contributed or collected data. We would also like to thank the research participants and employees of 23andMe for making this work possible. This work is supported by grants NIH-F32DK102323, NIH-4T32HL007901, NIH-R01DK107859, NIH-R35HL135818, NIH-K23DK114551 (M.S.U.), MGH Research Scholar Fund, Academy of Finland #309643 (H.M.O.), Instrumentarium Science Foundation (H.M.O.), Yrjö Jahnsson Foundation (H.M.O.), and Medical Research Council grant: MR/M005070/1. This work has been supported in part by The Spanish Government of Investigation, Development and Innovation (SAF2017-84135-R) including FEDER co-funding; The Autonomous Community of the Region of Murcia through the Seneca Foundation (20795/PI/18) and NIDDK R01DK105072 granted to M.G. The MEGASTROKE project received funding from sources specified at http://www.megastroke.org/acknowledgments.html.

## Author contributions

The study was designed by H.S.D., I.D., M.G., and R.S. H.S.D., I.D., J.M.L., Y.H., M.S.U., H.W., H.M.O., S.E.J., J.K., A.R.W., M.N.W., S.A., M.G., and R.S. participated in acquisition, analysis, and/or interpretation of data. H.S.D., I.D., M.G., and R.S. wrote the manuscript and all co-authors reviewed and edited the manuscript, before approving its submission. R.S. is the guarantor of this work and, as such, had full access to all the data in the study and takes responsibility for the integrity of the data and the accuracy of the data analysis.

## Competing interests

Y.H., S.A., and members of the 23andMe Research Team are employed by and hold stock or stock options in 23andMe, Inc. All remaining authors declare no competing interests.

## Additional information

**23andMe Research Team**

Michelle Agee[4], Adam Auton[4], Robert K. Bell[4], Katarzyna Bryc[4], Sarah K. Clark[4], Sarah L. Elson[4], Kipper Fletez-Brant[4], Pierre Fontanillas[4], Nicholas A. Furlotte[4], Pooja M. Gandhi[4], Karl Heilbron[4], Barry Hicks[4], David A. Hinds[4], Karen E. Huber[4], Ethan M. Jewett[4], Yunxuan Jiang[4], Aaron Kleinman[4], Keng-Han Lin[4], Nadia K. Litterman[4], Marie K. Luff[4], Jennifer C. McCreight[4], Matthew H. McIntyre[4], Kimberly F. McManus[4], Joanna L. Mountain[4], Sahar V. Mozaffari[4], Priyanka Nandakumar[4], Elizabeth S. Noblin[4], Carrie A. M. Northover[4], Jared O'Connell[4], Aaron A. Petrakovitz[4], Steven J. Pitts[4], G. David Poznik[4], J. Fah Sathirapongsasuti[4], Anjali J. Shastri[4], Janie F. Shelton[4], Suyash Shringarpure[4], Chao Tian[4], Joyce Y. Tung[4], Robert J. Tunney[4], Vladimir Vacic[4], Xin Wang[4] & Amir S. Zare[4]

