## [Peer Review File · Nature Communications]

REVIEWER COMMENTS

Reviewer #1 (Remarks to the Author):

The manuscript by Dashti et al represents genome-wide analyses of daytime napping in a large sample size. Such a large-scale study would be important to elucidate the genetic background of daytime napping. The manuscript is well-written, and the study methods are sound. However, I have several concerns as follows.

1 GWAS and other analyses adjusting for BMI were conducted. As a result, 110 of the 123 loci retained genome-wide significance. I have a concern about the adjustment of BMI. Is there a possibility that the adjustments for only BMI can be rather too simple?

I have read a GWAS paper of obstructive sleep apnea (OSA)-related traits (10.1164/rccm.201512-2431OC). In the OSA GWAS, analyses were adjusted for age, age × age, sex, age × sex, BMI, and BMI × BMI to find independent loci of obesity.

Not only sleep apnea and but also obesity would be associated with increasing daytime napping. My concern is that the effects of BMI or obesity have not been appropriately removed in the analyses. For example, the associations with 110 loci might be affected by the effect of BMI × BMI or obesity. Other covariates, such as BMI × BMI, should be investigated.

Sleep apnea is a complex disorder, and is strongly associated with obesity. Previous cohort studies have reported that the prevalence of OSA is a mean of 22% (range: 9-37%) in men and 17% (range: 4-50%) in women, therefore sleep apnea is a common disease. How many participants were diagnosed (or estimated) with sleep apnea? Are participants taking frequent naps likely to be sleep apnea? Sleep apnea increases blood pressure. How did the authors consider the effect of sleep apnea on results in the present study? Discussion about sleep apnea is also needed.

Complex associations among BMI, obesity and sleep apnea might be associated with daytime napping. Its complexity makes it difficult to understand the significance of the information I receive. To eliminate the above concern, would you reply to my comment?

2 Regarding mendelian randomization analyses, a previous report showed a causal effect of BMI on daytime sleepiness (doi.org/10.1038/s41467-019-11456-7), while genetically instrumented BMI was not associated with daytime napping in the current study. However, a shared genetic correlation between these two similar traits was observed ($r = 0.70$). I would like to know the reason of the difference or discrepancy between the two results.

3 Missense variants in HCRTR1, HCRTR2, TRPC6 and so on were identified to be primarily associated with daytime napping. Were other missense variants including rare variants in these genes investigated?

I looked at variants in HCRTR2 using The Sleep Disorder Knowledge Portal (<http://sleepdisordergenetics.org/home/portalHome>). p.P11T (rs41271312) and p.P10S (rs41271310) were nominally associated with daytime napping ($P = 0.00260$ and $P = 0.00130$, respectively). However, I don't know whether the result was adjusted by the effect of p.I308V (rs2653349) or not. There are several missense variants in these genes. I recommend that the missense variants be listed in a supplementary table. Please include P values before and after the adjustment of each leading missense variant.

4 Data of several genetic variants (rs614987,,,,) which be shown by “-“ are missing in Supplementary Table 5. Does this mean not-replicated variants? Data of not-replicated variants should be also shown.

5 Figure 4 and Supplementary Table 19 show a significant genetic correlation between daytime napping and psychiatric traits. This correlation is intriguing. Were mendelian randomization analyses with psychiatric traits and daytime napping performed? I would suggest to discuss the association between daytime napping and psychiatric traits in this manuscript.

6 I don't follow the logic about a paragraph of “leveraging HCRTR1 and HCRTR2 genetic associations to assess the cardiovascular safety profile of dual orexin antagonists”. Two missense variants in HCRTR1 and 2 were not associated with cardiovascular outcomes. This is a genetic association between variants and phenotypes. Pharmacological research would be required to evaluate adverse side effects of drugs. Overall, the discussion and conclusion would be overstated.

7 Minor comment: lines 774-776, an analogous approach was taken for reverse MR of adiposity measures (waist circumference, waist-hip-ratio adjusted for BMI, BMI) on daytime napping. Is “(waist circumference, WHRadjBMI and BMI)” better?

Another comment: The authors previously performed a GWAS of excessive daytime sleepiness from the UK biobank data. In the present study, the authors report a GWAS of daytime napping using the UK biobank data. Excessive daytime sleepiness and daytime napping are similar traits. In fact, the present study shows a significant genetic correlation between the two traits in Supplementary Table 19. Could not the authors have integrated both GWASs into one report? Could you tell the reviewer

why the authors split the GWASs into two papers? Actually, the authors previously reported GWASs of several sleep disturbance traits into a paper (doi.org/10.1038/ng.3749).

In addition, why did the GWAS of napping identify more loci than that of excessive daytime sleepiness (123 loci vs. 42 loci)? To begin with, what is the difference between excessive daytime sleepiness and daytime napping?

Reviewer #2 (Remarks to the Author):

Dashti et al performed a genome-wide search for genetic causes of daytime napping using large datasets from UK Biobank and 23&me, identifying 123 loci in the UK Biobank with strong evidence of replication in 23&me. The authors claim the associations retained significance when GWAS was restricted to healthy participants (see below, however). As the authors state in their Discussion, the results i) advance the biology of daytime napping by providing unprecedented insight in its genetic basis, ii) refine the understanding of pleiotropy and causality in the relationship with sleep and cardiometabolic traits (suggesting that daytime napping is rather the cause than the consequence of cardiometabolic disorders), and iii) inform pharmacologic investigations of orexin antagonism (by phenomic assessment of HCRTR missense mutations).

The authors use up-to-date methods for their data analyses (e.g., multi-trait clustering suggested the possibility of at least three heterogeneous pathways influencing daytime napping). The manuscript conveys a complex set of results:

Criticisms

1) [lines 101-102, Suppl. Table 1] The authors included a GWAS restricted to 338,764 out of 452,633 UKB participants, who are self-reporting excellent or good overall health. However, health indicators (BMI, waist circumference, blood pressure) differed minimally (<2.5%) in that subset of UKB participants as can be calculated from Suppl. Table 1 (i.e., these indicators also are in the pathological range). Hence, is it really true that those individuals that report good health actually are in good health? Substantial differences can be found only for smoking status (6.4% more frequent in those who report to be in excellent or good health) and for the Townsend social deprivation index (18.1% lower). Thus, substantial differences are given by more smoking and especially by less social deprivation (i.e., higher standard of living). Hence, the real difference of those self-reporting good health might relate to neglect of health issues and/or to more educated self-assessment.

2) [Suppl. Table 1] On average, the Townsend social deprivation index in the UKB discovery sample is 1.5 standard deviations below the mean of 0, as can be calculated from Suppl. Table 1). Hence the UKB participants are substantially less deprived (that is, more affluent) than the UK population average. Therefore, the UKB sample cannot be regarded as a representative population sample. The authors might want to comment on that deviation and how it potentially impacts their analyses (e.g., when performing genetic correlation analysis with educational achievement).

3) [lines 286 ff] Using Mendelian randomization analyses the authors suggest that daytime napping is rather the cause than the consequence of cardiometabolic traits. Questions: i) How do they exclude the possibility that the true cause is not daytime napping itself but some other sleep trait (e.g. insomnia) with which daytime napping is genetically correlated? ii) Is it possible that the obvious genetic heterogeneity of daytime napping interferes with an adequate application of Mendelian randomization? (Would MR analyses of the individual clusters versus daytime napping provide the same direction of causation?)

4) [Suppl. Tables 1 and 2, lines 104-108] The authors report a highly significant difference in napping frequency between males and females (with the latter napping much less; $p < 0.001$, see Suppl. Table 1) but do not find any evidence of sex heterogeneity between their sex-specific GWAS signals (see Suppl. Table 2) or in their genetic correlation analysis between males and females (see lines 104-108). They might want to comment on that. Is there a major influence of the X chromosome or is there a life history explanation of the phenotypic napping difference between the sexes?

5) [lines 130-134] The authors state, “we further quantified the impact of daytime napping on daytime inactivity duration using a polygenic score comprised of lead variants at all 123 loci. A category increase in frequency of daytime napping was associated with 18.9 minutes (95% confidence interval =13.6, 24.2; $P = 4.21 \times 10^{-12}$) longer duration of daytime inactivity.” Why is it necessary to take the detour via genetics (using the polygenic score) to establish the association between frequency of napping and duration of napping? Why not comparing directly the phenotypic data?

6) [lines 132-140, 144-146] The authors state that “a category increase in frequency of daytime napping ... had no effect on other accelerometer-derived sleep duration, timing or quality phenotypes.” However, they also state that “individual daytime napping loci also associated with self-reported sleep traits and accelerometer-derived sleep measures.” Furthermore, they state that “overlap [with daytime napping] was also supported by cross-trait LD score regression where we observed strongest evidence for a shared genetic basis with daytime sleepiness ... and long sleep duration.” The first of these statements is at odds with the latter two. Is there a (genetic) association (“overlap”) between daytime napping and sleep duration, or is there not?

7) [lines 370-371] The authors state that daytime napping shows genetic correlations with HDL cholesterol and lipid subfractions suggesting uncharacterized pleiotropic roles of lipid metabolism on sleep. Are these correlations still significant after correction for BMI?

8) [Suppl. Tables 19] The significances of the genetic correlation analyses were Bonferroni-corrected for the number of all tests performed ($P < 0.05/257$ tests). This might be too conservative because the traits included in the analyses are correlated. Insomnia, for instance, is included twice (from different analyses of the same UKB data).

9) [Suppl. Table 2, lines 235-236] The authors identified the BTBD9 locus in their GWAS on daytime napping but do not further comment on it. Neither do they indicate that the same locus (albeit with different lead SNPs) has been identified prominently in GWASs on other sleep traits, i.e., on periodic leg movements (PLMs; Stefansson et al 2007, N Engl J Med 2007; 357:639-647), on RLS (Schormair et al, Lancet Neurology 2017), and on insomnia symptoms (Jansen et al, Nat Genet 2019; 51:394-403). Indeed, the lead SNP rs3923809 of the PLMs GWAS is highly correlated with lead SNP rs4236060 of the GWAS on daytime napping reported in the present manuscript. The authors might want to comment on that, especially because BTBD9 is a prominent member of their cluster “early sleep timing”.

10) [lines 167-170] The authors report a missense variant “in a transmembrane helical domain of HCRTR2 [I308V;rs2653349; A effect allele frequency (EAF)=0.21; associated with more frequent daytime napping, morning preference and ease of awakening, posterior probability of colocalization (pp)=0.98] reflecting putative loss of function”. It is not sufficiently clear how they interfere this “putative loss of function”.

11) [lines 164ff, Suppl. Table 11] For the potentially causative variants identified by colocalization analysis listed in Suppl. Table 11 (and partially discussed in the main text) it would be helpful to add the CADD scores, etc..

12) [lines 180-185] The following sentence is not sufficiently clear. “As further evidence that two distinct signals drive the daytime sleepiness and daytime napping associations, we conducted conditional analysis using GCTA COJO adjusting the daytime napping association for the lead daytime napping variant (rs2653349), and observed attenuation in the association for the lead daytime sleepiness variant in HCRTR2 (rs3122170) (P value from 4.60×10^{-18} to 4.56×10^{-3}).” If the signals are independent, conditional analysis should NOT substantially change the p-values.

Reviewer #3 (Remarks to the Author):

Authors have performed gwas on day time napping in UKB and 23andME. They identified 123 loci of which 60 were replicated. Authors identify association with hypertension

and waist circumference and further identify causal links.

The manuscript is well written and analyses are performed carefully. I have a couple of issues:

1- The accelerometer provides an objective assessment of day time inactivity/napping. If the associations were true they should have more power to be detected with accelerometer data but

what you see is only a few variants appear associated with the accelerometer based assessment. Would the authors comment on why this is?

2- In their MR analysis, authors mention that SBP and DBP summary data are not available and therefore split the UKB into two sets. Have the authors checked with ICBP consortium. They have led the largest GWAS and summary statistics are also available upon request.

Reviewer #1 (Remarks to the Author):

The manuscript by Dashti et al represents genome-wide analyses of daytime napping in a large sample size. Such a large-scale study would be important to elucidate the genetic background of daytime napping. The manuscript is well-written, and the study methods are sound. However, I have several concerns as follows.

RESPONSE: We would like to thank the Reviewer for their thorough assessment of our work. We have addressed all comments from this Reviewer below.

1. GWAS and other analyses adjusting for BMI were conducted. As a result, 110 of the 123 loci retained genome-wide significance. I have a concern about the adjustment of BMI. Is there a possibility that the adjustments for only BMI can be rather too simple? I have read a GWAS paper of obstructive sleep apnea (OSA)-related traits (10.1164/rccm.201512-2431OC). In the OSA GWAS, analyses were adjusted for age, age \times age, sex, age \times sex, BMI, and BMI \times BMI to find independent loci of obesity. Not only sleep apnea and but also obesity would be associated with increasing daytime napping. My concern is that the effects of BMI or obesity have not been appropriately removed in the analyses. For example, the associations with 110 loci might be affected by the effect of BMI \times BMI or obesity. Other covariates, such as BMI \times BMI, should be investigated.

Sleep apnea is a complex disorder, and is strongly associated with obesity. Previous cohort studies have reported that the prevalence of OSA is a mean of 22% (range: 9-37%) in men and 17% (range: 4-50%) in women, therefore sleep apnea is a common disease. How many participants were diagnosed (or estimated) with sleep apnea? Are participants taking frequent naps likely to be sleep apnea? Sleep apnea increases blood pressure. How did the authors consider the effect of sleep apnea on results in the present study? Discussion about sleep apnea is also needed. Complex associations among BMI, obesity and sleep apnea might be associated with daytime napping. Its complexity makes it difficult to understand the significance of the information I receive. To eliminate the above concern, would you reply to my comment?

RESPONSE: We agree with the Reviewer that adjusting for BMI alone might be too simplistic. Thus, in recognizing the possibility that the adjustment for BMI alone might be insufficient, we now run an additional GWAS model adjusting for BMI and BMI \times BMI. Overall, findings for our GWAS signals remained largely unchanged. We observed no difference in the number of signals that retained GWAS significance (110 loci) upon further accounting for BMI \times BMI compared to BMI alone (see **figure**). This is not surprising as the inclusion of BMI \times BMI to the statistical model only explains an additional 0.02% of daytime napping variance. Results from the BMI \times BMI model have now been added to **Supplementary Table 2**.

We agree with the Reviewer that sleep apnea is a common and relevant disease to consider for our analysis. In our UK Biobank sample, the prevalence of diagnosed sleep apnea is 1.2% as determined from ICD-10 diagnosis code, and similar to other UK Biobank publications (PBMID: 32060260). Overall, the prevalence of diagnosed sleep apnea is higher with more frequent napping: 3.6%, 1.6%, and 0.8% for always, sometimes and never daytime napping, respectively. We partly account for diagnosed sleep apnea when we previously restricted our analysis to the “healthy UK Biobank subset,” where the prevalence of diagnosed sleep apnea decreased from 1.2% to 0.7%. As we’ve stated previously, findings for daytime napping in this “healthy UK Biobank subset” were largely similar to those in the larger UK Biobank sample.

To further carefully examine the influence of sleep apnea on our results, as suggested by the Reviewer, we now run two additional sets of GWAS:

a) GWAS **excluding all participants with diagnosed sleep apnea** in the UK Biobank ($n = 5,553$ excluded); and, b) GWAS with all UK Biobank participants **adjusted by a modified STOP-BANG risk scale** (reference 67). The modified STOP-BANG risk scale for sleep apnea is missing the question “Has anyone observed you stop breathing during sleep?” and replacing neck circumference with waist circumference dichotomized to the threshold for metabolic syndrome (reference 21). We have previously developed this risk scale for sleep apnea in the UK Biobank to account for undiagnosed sleep apnea and confirmed that patients with diagnosed sleep apnea have higher scores compared to controls.

Overall, we observe that 114 of the 123 GWAS signals retained significance after excluding diagnosed sleep apnea and 102 of the 123 signals remained significant after adjusting for the modified STOP-BANG. However, effect estimates were minimally attenuated (see **figures**).

We now include the prevalence of sleep apnea diagnosis in **Supplementary Table 1** and results from the two sensitivity GWAS runs in **Supplementary Table 2**. In addition, we further added in the Results the following (page 6, line 122): “Accounting for sleep apnea in GWAS models excluding participants with diagnosed sleep apnea ($n = 5,553$ excluded) or adjusting by a modified STOP-BANG risk scale²¹ did not influence findings (**Supplementary Table 2**).”

2. Regarding mendelian randomization analyses, a previous report showed a causal effect of BMI on daytime sleepiness (doi.org/10.1038/s41467-019-11456-7), while genetically instrumented BMI was not associated with daytime napping in the current study. However, a shared genetic correlation between these two similar traits was observed ($r = 0.70$). I would like to know the reason of the difference or discrepancy between the two results.

RESPONSE: We thank the reviewer for raising this interesting point. While it is true that the two traits are highly correlated, as noted by the Reviewer, our analyses show that there are several key differences in the heritability and loci identified for daytime napping and daytime sleepiness. For example, the SNP-based heritability of daytime napping (11.9%) was almost double that previously reported for daytime sleepiness (6.9%). Some of the variants identified for daytime napping are unique to this trait and not shared with other phenotypes. Phenotypically, we observed that among participants reporting *always* daytime napping, 51.8% also report *never/rarely* having daytime sleepiness. Thus considering that daytime napping and daytime sleepiness are correlated but distinct traits, it is unsurprising that there are differences in the GWAS findings and subsequently in the MR results.

We now include the distribution of daytime sleepiness in **Supplementary Table 1**.

We now elaborate upon these differences in our Discussion (page 18, lines 636): “Although daytime napping shares biological determinants with other sleep traits, most prominently daytime sleepiness²⁴, there were several genetic findings unique to daytime napping. There were 26/123 loci unique to daytime napping, with several other loci exhibiting stronger relationships with daytime napping relative to other traits (e.g. *KSR2* locus). The SNP-based heritability of daytime napping (11.9%) was almost double that previously reported for daytime sleepiness (6.9%)²⁴, and daytime napping variants were modestly attenuated in GWAS models accounting for daytime sleepiness. Although prior analyses related higher BMI to increased frequency of daytime sleepiness²⁴, we observed no such relationship with frequency of daytime napping. In contrast, we found detrimental effects of daytime napping on cardiometabolic health, which were not previously observed for daytime sleepiness. Taken together, these data suggest that daytime napping and daytime sleepiness should be considered related, but distinct features of the impaired arousal continuum.”

3. Missense variants in HCRTR1, HCRTR2, TRPC6 and so on were identified to be primarily associated with daytime napping. Were other missense variants including rare variants in these genes investigated? I looked at variants in HCRTR2 using The Sleep Disorder Knowledge Portal (<http://sleepdisordergenetics.org/home/portalHome>). p.P11T (rs41271312) and p.P10S (rs41271310) were nominally associated with daytime napping (P = 0.00260 and P= 0.00130, respectively). However, I don't know whether the result was adjusted by the effect of p.I308V (rs2653349) or not. There are several missense variants in these genes. I recommend that the missense variants be listed in a supplementary table. Please include P values before and after the adjustment of each leading missense variant.

RESPONSE: We thank the Reviewer for bringing these missense variants to our attention. To minimize false positive findings, in general we have used the conventionally accepted threshold for genome-wide significance ($P < 5 \times 10^{-8}$). We may not have adequate power to detect low frequency missense variants, such as rs41271310 in *HCRTR2* (MAF =0.003). As further clarification, none of the variant effects provided on the Portal are conditioned on the effect of any other variants.

For your reference, we provide below a table of nominally associated ($P < 0.05$) missense variant associations before and after conditioning on the lead missense variant in the region (there were no other missense variants in *HCRTR1*):

Gene	Tested SNP	Conditioned SNP	Chr	Pos	EA	EAF	Unconditioned Effect			Conditioned Effect		
							Beta	SE	P value	Beta	SE	P value
HCRTR2	rs41271310	rs2653349	6	55039413	C	0.997	-0.034	0.011	1.32E-03	-0.036	0.011	5.42E-04
HCRTR2	rs41271312	rs2653349	6	55039416	C	0.991	-0.019	0.006	3.26E-03	-0.022	0.006	6.08E-04
TRPC6	rs36111323	rs3802829	11	101359750	G	0.879	0.006	0.002	2.16E-03	0.003	0.002	1.45E-01

Chr: chromosome; EA: effect allele for tested SNP; EAF: effect allele frequency of tested SNP; Pos: position of tested SNP; SE: standard error; SNP: single nucleotide polymorphism.

4. Data of several genetic variants (rs614987,,,) which be shown by “-“ are missing in Supplementary Table 5. Does this mean not-replicated variants? Data of not-replicated variants should be also shown.

RESPONSE: In Supplementary Table 6 (previously Supplementary Table 5), genetic variants previously shown with “-“ results indicated missing data from 23andMe and/or UK Biobank+23andMe meta-analysis. All data for replicated and non-replicated variants from 23andMe and the UK Biobank+23andMe meta-analysis are shown in Supplementary Table 6. We now replace “-“ with NA and indicate in the footnote that NA reflects missing data on genetic variant.

5. Figure 4 and Supplementary Table 19 show a significant genetic correlation between daytime napping and psychiatric traits. This correlation is intriguing. Were mendelian randomization analyses with psychiatric traits and daytime napping performed? I would suggest to discuss the association between daytime napping and psychiatric traits in this manuscript.

RESPONSE: Our manuscript is limited in scope to cardiometabolic health outcomes, as reflected in the title, and therefore, we have not investigated in MR potential causal links with psychiatric traits.

We now remove psychiatric traits from Figure 4 to further emphasize our cardiometabolic results. In addition, we now mention that the focus of our paper on cardiometabolic traits as a limitation (page 22, line 784): “In addition, our analysis was limited in scope to cardiometabolic health, and future studies should evaluate the impact of daytime napping on other health outcomes including mental health.”

6. I don't follow the logic about a paragraph of “leveraging HCRTR1 and HCRTR2 genetic associations to assess the cardiovascular safety profile of dual orexin antagonists”. Two missense variants in HCRTR1 and 2 were not associated with cardiovascular outcomes. This is a genetic association between variants and phenotypes. Pharmacological research would be required to evaluate adverse side effects of drugs. Overall, the discussion and conclusion would be overstated.

RESPONSE: The reviewer makes a valid point and we have modified the text in the Discussion to clarify the notion that we are using genetics to predict clinical effects, rather than assessing the effects of the drugs themselves (which, as the reviewer states, would require pharmacological research).

We have modified the header of this section in the Results (page 15, line 488): *Leveraging HCRTR1 and HCRTR2 genetic associations to **predict** the cardiovascular safety profile of dual orexin antagonists*”

In the Discussion we state (page 20, line 717): “We leveraged coding variation in *HCRTR1* and *HCRTR2* to predict the cardiovascular consequences of long-term pharmacologic modulation of orexin receptors. We found no net effect of these genetic proxies on cardiovascular outcomes, nor on any ICD-code defined disease outcomes in a phenome-wide association study. These results predict that pharmacologic agonism or antagonism of orexin receptors therapies is unlikely to increase risk of cardiovascular disease. A novel association of *HCRTR1* and *HCRTR2* with blood pressure was observed, however the direction of effect differed for the two variants. This suggests a neutral net blood pressure effect of dual orexin receptor antagonism, and more broadly suggests pleiotropic effects of these proteins on blood pressure regulation. However, it is possible that these genetic variants do not proxy peripheral effects of *HCRTR1* and *HCRTR2* inhibition (e.g. bone marrow)⁴⁹. This is the first application of PheWAS to study on-target side effects of sleep medications, and sets the stage for future use of these genetic proxies to understand the health consequences of orexin receptor modulation.”

7. Minor comment: lines 774-776, an analogous approach was taken for reverse MR of adiposity measures (waist circumference, waist-hip-ratio adjusted for BMI, BMI) on daytime napping. Is “(waist circumference, WHRadjBMI and BMI)” better?

RESPONSE: We have replaced **waist-hip-ratio adjusted for BMI** with **WHRadjBMI** per the Reviewer’s suggestion in the text (page 38, line 1191).

8. The authors previously performed a GWAS of excessive daytime sleepiness from the UK biobank data. In the present study, the authors report a GWAS of daytime napping using the UK biobank data. Excessive daytime sleepiness and daytime napping are similar traits. In fact, the present study shows a significant genetic correlation between the two traits in Supplementary Table 19. Could not the authors have integrated both GWASs into one report? Could you tell the reviewer why the authors split the GWASs into two papers? Actually, the authors previously reported GWASs of several sleep disturbance traits into a paper (doi.org/10.1038/ng.3749).

RESPONSE: As per our previous response to comment 2 above, we believe while related, these traits are different enough so as to warrant separate, in-depth studies. While daytime napping is correlated to other sleep traits we have previously investigated, including daytime sleepiness, we are learning that each sleep trait is unique. Therefore, GWAS results from these investigations only partly overlap, but are largely specific to the trait being investigated.

We have decided not to integrate findings from both GWASs into a single report because our approach for each investigation is different and in order to allow thorough examination and description of study results. For example, the GWAS for daytime napping is focused on cardiometabolic outcomes, utilizes the 23andMe cohort for replication, and leverages the Mass General Brigham Biobank for PheWAS. Our previously published multi-trait GWAS for sleep disturbance did include daytime napping, however due to the limited overall GWAS findings for each trait, limited methodology, and modest sample sizes, summarizing results into a single report was feasible.

9. In addition, why did the GWAS of napping identify more loci than that of excessive daytime sleepiness (123 loci vs. 42 loci)? To begin with, what is the difference between excessive daytime sleepiness and daytime napping?

RESPONSE: The third edition of the International Classification of Sleep Disorders defines excessive daytime sleepiness as the “inability to maintain wakefulness and alertness during the major waking episodes of the day, with sleep occurring unintentionally or at inappropriate times almost daily for at least three months” (1). Although the daytime napping phenotype we studied may occur in the setting of daytime sleepiness (as evidenced by the high genetic correlation between the phenotypes), the distinct features of the genetic architecture of daytime napping point to unique biology (*see response to comment 2 above*).

As expected, observe higher prevalence of daytime sleepiness with more frequent napping. However, among participants who report *always* daytime napping, 51.8% of participants also report *never/rarely* having daytime sleepiness. These phenotypic findings suggest that daytime napping and daytime sleepiness are related, but distinct phenotypes.

In addition, we now run an additional GWAS model further adjusting for daytime sleepiness for sensitivity. We observe modest attenuation of effect estimates after adjusting for daytime sleepiness, and that 60 of the 123 signals retained GWAS significance after adjusting for daytime sleepiness.

These new GWAS findings, along with the previously observed difference in trait heritability estimates further suggesting that daytime napping and daytime sleepiness are related, but distinct phenotypes with unique genetic architectures.

We now include the distribution of daytime sleepiness in **Supplementary Table 1**, and sensitivity GWAS findings from the analytical model accounting for daytime sleepiness in **Supplementary Table 1**.

In the Results, we add the following (page 6, line 124): “Finally, when adjusting for daytime sleepiness, we observe modest attenuation of effect estimates, with 60 of the 123 loci retaining genome-wide significance (**Supplementary Table 2**).”

In the Discussion, we add the following (page 19, line 670): “The SNP-based heritability of daytime napping (11.9%) was almost double that previously reported for daytime sleepiness (6.9%)²⁴, and daytime napping variants were modestly attenuated in GWAS models accounting for daytime sleepiness... Taken together, these data suggest that daytime napping and daytime sleepiness should be considered related, but distinct features of the impaired arousal continuum.”

Reference:

1. Chervin R. Approach to the patient with excessive daytime sleepiness. UpToDate. 2019

Reviewer #2 (Remarks to the Author):

Dashti et al performed a genome-wide search for genetic causes of daytime napping using large datasets from UK Biobank and 23&me, identifying 123 loci in the UK Biobank with strong evidence of replication in 23&me. The authors claim the associations retained significance when GWAS was restricted to healthy participants (see below, however). As the authors state in their Discussion, the results i) advance the biology of daytime napping by providing unprecedented insight in its genetic basis, ii) refine the understanding of pleiotropy and causality in the relationship with sleep and cardiometabolic traits (suggesting that daytime napping is rather the cause than the consequence of cardiometabolic disorders), and iii) inform pharmacologic investigations of orexin antagonism (by phenomic assessment of HCRTR missense mutations).

The authors use up-to-date methods for their data analyses (e.g., multi-trait clustering suggested the possibility of at least three heterogeneous pathways influencing daytime napping). The manuscript conveys a complex set of results:

1. [lines 101-102, Suppl.Table 1] The authors included a GWAS restricted to 338,764 out of 452,633 UKB participants, who are self-reporting excellent or good overall health. However, health indicators (BMI, waist circumference, blood pressure) differed minimally (<2.5%) in that subset of UKB participants as can be calculated from Suppl. Table 1 (i.e., these indicators also are in the pathological range). Hence, is it really true that those individuals that report good health actually are in good health? Substantial differences can be found only for smoking status (6.4% more frequent in those who report to be in excellent or good health) and for the Townsend social deprivation index (18.1% lower). Thus, substantial differences are given by more smoking and especially by less social deprivation (i.e., higher standard of living). Hence, the real difference of those self-reporting good health might relate to neglect of health issues and/or to more educated self-assessment.

RESPONSE: In earlier assessments in the UK Biobank, it was observed that measures obtained by questionnaires and without physical examination were the strongest predictors of all-cause mortality in that population (reference 51). Specifically, self-reported overall health rating, which was used in the present analysis, was the strongest predictor for mortality in men and 4th strongest predictor for mortality in women, and outperformed 655 other measures. This suggests that this self-reported measure is likely capturing a substantial burden of objective poor health. Therefore, excluding participants with poor self-reported health rating is an appropriate approach to examine whether the detected genetic variants are driven by poor health. In addition, and as acknowledged by this Reviewer in comment 2 below, the UK Biobank is regarded as an overall healthy population.

We now emphasize this in the Discussion (page 17, line 584): “Variant effects were largely independent of BMI and sleep apnea, and the associations retained significance when GWAS was restricted to healthier participants, a strong determinant of 5 year mortality in the UK Biobank⁵¹, suggesting that signals were not driven by poor health.”

2. [Suppl. Table 1] On average, the Townsend social deprivation index in the UKB discovery sample is 1.5 standard deviations below the mean of 0, as can be calculated from Suppl. Table 1). Hence the UKB participants are substantially less deprived (that is, more affluent) than the UK population average. Therefore, the UKB sample cannot be regarded as a representative population sample. The authors might want to comment on that deviation and how it potentially impacts their analyses (e.g., when performing genetic correlation analysis with educational achievement).

RESPONSE: We agree with the Reviewer that the UK Biobank cannot be regarded as a representative population sample. The low participation rate and relative healthiness of the UK Biobank at 5.5% may have introduced selection bias. Therefore, consistency of the genetic signals between the UK Biobank and 23andMe, an independent study with different demographic, and various findings with the Mass General Brigham Biobank, an independent clinical cohort, reflects the generalizability of our findings. Furthermore, several variants we identified are in pathways with known relevance to sleep (e.g. *HCRTR1* and *HCRTR2*), which suggests that we are capturing a true biological signal. Nonetheless, continued evaluation in other demographics, including age-groups and ancestries, is necessary.

We have added this to the Discussion (page 21, line 766): “The low participation rate of the UK Biobank at 5.5% may have introduced selection bias. However, consistency of the genetic signals between the UK Biobank and 23andMe, an independent study with different demographic, and various findings with the Mass General Brigham Biobank, an independent clinical cohort, supports the generalizability of our findings. In addition, the identification of variants in pathways with known relevance to sleep (e.g. *HCRTR1* and *HCRTR2*) suggests that the GWAS is capturing true biological signal. Nonetheless, continued evaluation in other demographics, including age-groups and ancestries, is necessary.”

3. [lines 286 ff] Using Mendelian randomization analyses the authors suggest that daytime napping is rather the cause than the consequence of cardiometabolic traits. Questions: i) How do they exclude the possibility that the true cause is not daytime napping itself but some other sleep trait (e.g. insomnia) with which daytime napping is genetically correlated? ii) Is it possible that the obvious genetic heterogeneity of daytime napping interferes with an adequate application of Mendelian randomization? (Would MR analyses of the individual clusters versus daytime napping provide the same direction of causation?)

RESPONSE: The reviewer makes a valid point that pleiotropy with other sleep traits may bias the MR estimates reported in our manuscript. To address this concern, we have now performed multivariable Mendelian randomization analyses adjusting for insomnia (as the reviewer noted), as well as sleep duration (given prior work from our group demonstrating causal effects of sleep duration on cardiovascular health).

The multivariable MR results are presented in **Supplementary Table 24** and the MR results were effectively unchanged.

We agree with the reviewer that heterogeneity limits inference from our Mendelian randomization analyses. In the Discussion we now state (page 22, line 790): “Further dissection of the heterogeneity of daytime napping is necessary to determine which types of daytime napping behavior are most detrimental to cardiometabolic health.”

We considered performing cluster-specific Mendelian randomization analyses, as suggested by the Reviewer. Many of the cluster-specific loci are pleiotropic with important cardiometabolic traits and risk factors (e.g. *PATJ* and BMI; *MTNR1B* and Type 2 Diabetes), which would necessitate the use of pleiotropy-robust methods for valid inference. Considering the limited number of variants within each cluster, there are very few variants available for such MR analysis after data harmonization (≤ 5 SNPs). Instead, we performed ‘cluster-specific polygenic score associations’, which are analogous to a fixed-effects inverse-variance weighted MR analysis. Results from these cluster-specific polygenic scores are presented in Table 1 and in the main text (page 14, line 437): “We also observed associations of a polygenic score of the 123 napping variants, and polygenic sub-scores for each of the 3 clusters with cardiometabolic traits from large-scale public GWAS (**Table 1, Supplementary Table 22**). Cluster-specific polygenic score associations varied across outcomes, and included associations of cluster 1 with higher blood pressure for cluster 1, and clusters 2 and 3 with adiposity traits (**Table 1**).”

4. [Suppl. Tables 1 and 2, liens 104-108] The authors report a highly significant difference in napping frequency between males and females (with the latter napping much less; $p < 0.001$, see Suppl. Table 1) but do not find any evidence of sex heterogeneity between their sex-specific GWAS signals (see Suppl. Table 2) or in their genetic correlation analysis between males and females (see lines 104-108). They might want to comment on that. Is there a major influence of the X chromosome or is there a life history explanation of the phenotypic napping difference between the sexes?

RESPONSE: Indeed, we observe that daytime napping is more prevalent among men compared to women, as has been previously reported in other nationally representative samples (reference 52). In further analysis, we observe that the sex-difference in daytime napping prevalence is present and increases throughout life history (i.e., difference in prevalence = 0.7% at 40 yrs vs. 2.0% at 50 yrs vs. 5.9% at 60 yrs vs. 8.6% at 70 yrs – see **figure**).

To further examine whether other common genetic factors contribute to sex differences in daytime napping, we now conduct an X chromosome association analysis. On the X chromosome, we identified 5 additional loci associated with daytime napping (**Supplementary Table 3**). One signal (rs6621715) had significantly different effects between the sexes ($P = 0.006$) but overall a modest effect on daytime napping frequency. It's unclear whether the difference in sex-stratified GWAS is due to power differences between the male and female subsamples or a true sex-specific genetic effect. No additional sex-specific GWAS variants

were identified on the X chromosome. Therefore, our results suggest that there is no major influence of the X chromosome on difference in daytime napping.

Thus, sex heterogeneity is unlikely due to autosomal common variants, or common variants on the X chromosome and future analyses investigating this sex heterogeneity is warranted. We now add in the Methods details on the X chromosome analysis (page 26, line 900): “X-chromosome data were imputed and analyzed separately using the same analytical approach in BOLT-LMM as was done for analysis of autosomes. A rare chrX signal at *IGSF1* on chromosome-X driven by one rare variant (rs189568347; MAF =0.006) was identified, potentially attributed to genotyping artifact or false-positive association and therefore was excluded.”

Findings from the X chromosome analysis are added in the Results (page 6, line 132): “We conducted association analyses on the X chromosome to further examine whether common variants on the X chromosome contribute to sex differences in daytime napping and identified 5 additional loci for daytime napping (**Supplementary Table 3**). Only one of these variants (rs6621715) had significantly different effect estimates in males and females ($P = 0.006$), and no additional GWAS signals were identified on the X chromosome in sex-stratified analysis.”

In the Discussion, we add the following (page 17, line 588): “In addition, despite higher prevalence of daytime napping among men compared to women⁵², we identified only one sex-specific signal on the X chromosome, suggesting sex differences may be attributed to environmental factors or possibly rare genetic variants.” Also, in the Discussion (page 22, line 792): “In addition, future analyses investigating sex heterogeneity in daytime napping frequency is warranted.”

5. [lines 130-134] The authors state, “we further quantified the impact of daytime napping on daytime inactivity duration using a polygenic score comprised of lead variants at all 123 loci. A category increase in frequency of daytime napping was associated with 18.9 minutes (95% confidence interval =13.6, 24.2; $P = 4.21 \times 10^{-12}$) longer duration of daytime inactivity.” Why is it necessary to take the detour via genetics (using the polygenic score) to establish the association between frequency of napping and duration of napping? Why not comparing directly the phenotypic data?

RESPONSE: The purpose of quantifying the impact of daytime napping on daytime inactivity duration from accelerometer is to partly validate the specificity of our discovered loci with an objectively determined daytime napping behavior. This exploration is important as it partly accounts for limitations of self-reported data. In addition, by exploring associations with other accelerometer-derived sleep traits, we are able to understand underlying physiologic mechanisms related to our traits. For example, since the polygenic score associated with duration of daytime inactivity but not other accelerometer-derived sleep duration, timing or quality phenotypes (as indicated in the Results) supports the specificity of our loci. Of note, the phenotypic correlation between these two measures is: Pearson correlation $r^2 = 0.17$.

We now clarify the purpose of our approach in the Results (page 7, line 168): “Given inherent limitations of self-reported data, we aimed to partly validate the specificity of our associations

with an objective measure corresponding to daytime napping behavior. We thus compared effect estimates of the 123 loci with effect estimates for accelerometer-derived daytime inactivity duration¹⁹ from 7-day wrist accelerometry obtained in 85,499 participants of European ancestry in the UK Biobank >2 years after baseline assessment.”

6. [lines 132-140, 144-146] The authors state that “a category increase in frequency of daytime napping ... had no effect on other accelerometer-derived sleep duration, timing or quality phenotypes.” However, they also state that “individual daytime napping loci also associated with self-reported sleep traits and accelerometer-derived sleep measures.” Furthermore, they state that “overlap [with daytime napping] was also supported by cross-trait LD score regression where we observed strongest evidence for a shared genetic basis with daytime sleepiness ... and long sleep duration.” The first of these statements is at odds with the latter two. Is there a (genetic) association (“overlap”) between daytime napping and sleep duration, or is there not?

RESPONSE: The reviewer raises a valid concern about the clarity of this section. Although there are correlations with subjective sleep traits, there are no polygenic risk score associations or genetic correlations with accelerometer-defined sleep duration. We emphasize that it is not surprising to observe a few pleiotropic loci with a null overall genetic correlation or polygenic risk score analysis.

We have now revised and rearranged this section of the manuscript to improve clarity in the Results (page 8, line 203): “Several daytime napping-associated variants had pleiotropic associations with other self-reported sleep traits^{23–26} and accelerometer-derived sleep measures¹⁹ (**Supplementary Table 7, 9**). This genetic overlap between daytime napping and other sleep traits was further supported by cross-trait LD score regression²⁷ where we observed the strongest evidence for a shared genetic basis with daytime sleepiness ($r_g = 0.70$, $P = 7.94 \times 10^{-373}$) and long sleep duration ($r_g = 0.42$, $P = 1.94 \times 10^{-64}$), and weaker correlations with other sleep duration, timing and quality phenotypes (**Supplementary Table 10**). In concordance with the null polygenic risk score association, daytime napping was not genetically correlated with accelerometer-defined sleep duration. Despite the observed genome-wide genetic overlap, lead variants at 26 of the 123 loci showed no statistical evidence for association with previously studied sleep traits in the UK Biobank ($P_{adj} > 0.05$), suggesting that these variants reflect mechanisms specific to daytime napping (**Supplementary Table 9**).”

7. [lines 370-371] The authors state that daytime napping shows genetic correlations with HDL cholesterol and lipid subfractions suggesting uncharacterized pleiotropic roles of lipid metabolism on sleep. Are these correlations still significant after correction for BMI?

RESPONSE: In Supplementary Table 20, we present daytime napping genome-wide genetic correlations with HDL cholesterol and lipid subfractions from models without and with BMI adjustment. Indeed, only correlations with triglycerides remain significant when accounting for BMI.

We now highlight FDR significant genetic correlations in Figure 4a with an asterisk. In the text, we indicate that correlations, except for triglycerides, were no longer significant after accounting for BMI in the GWAS model (page 13, line 411): “Modest positive correlations were observed between daytime napping and several anthropometric and cardiometabolic diseases and traits including BMI, triglycerides, and type 2 diabetes (**Figure 4a, Supplementary Table 20**), of which correlations with triglycerides remained significant in the GWAS model adjusting for BMI.”

8. [Suppl. Tables 19] The significances of the genetic correlation analyses were Bonferroni-corrected for the number of all tests performed ($P < 0.05/257$ tests). This might be too conservative because the traits included in the analyses are correlated. Insomnia, for instance, is included twice (from different analyses of the same UKB data).

RESPONSE: In Supplementary Table 20 (previously Supplementary Table 19), we now present unadjusted, FDR corrected, and Bonferroni corrected P values since some of the traits are correlated as indicated by the Reviewer. Compared to the more stringent Bonferroni corrected P values, we observed 69 additional correlations with the less stringent FDR correction, including several anthropometric and lipid traits. We now indicate FDR significant correlations in Figure 4 with an asterisk.

9. [Suppl. Table 2, lines 235-236] The authors identified the BTBD9 locus in their GWAS on daytime napping but do not further comment on it. Neither do they indicate that the same locus (albeit with different lead SNPs) has been identified prominently in GWASs on other sleep traits, i.e., on periodic leg movements (PLMs; Stefansson et al 2007, N Engl J Med 2007; 357:639-647), on RLS (Schormair et al, Lancet Neurology 2017), and on insomnia symptoms (Jansen et al, Nat Genet 2019; 51:394-403). Indeed, the lead SNP rs3923809 (0.85) of the PLMs GWAS is highly correlated with lead SNP rs4236060 of the GWAS on daytime napping reported in the present manuscript. The authors might want to comment on that, especially because BTBD9 is a prominent member of their cluster “early sleep timing”.

RESPONSE: We agree with the reviewer that the pleiotropic association of this variant with PLMs and with napping should be highlighted in the manuscript. We now state (page 11, line 327): “Fourth, several genetic variants were prioritized at or near genes a) coding for proteins constituting or interacting with potassium channels [rs77154532 (*KCHN8*), rs10875606 (*KCTD16*)], b) involved in glutamate transmission [rs60920123 (*GRIN2A*), rs2284015 (*CACNG2*)], and c) previously associated with periodic leg movements³⁷ and restless legs syndrome³⁸ [rs4236060 (*BTBD9*)].”

10. [lines 167-170] The authors report a missense variant “in a transmembrane helical domain of HCRTR2 [I308V;rs2653349; A effect allele frequency (EAF)=0.21; associated with more frequent daytime napping, morning preference and ease of awakening, posterior probability of colocalization (pp)=0.98] reflecting putative loss of function”. It is not sufficiently clear how they infer this “putative loss of function”.

RESPONSE: The Reviewer makes a valid point. We now remove mention of “putative loss of function” from the text.

11. [lines 164ff, Suppl. Table 11] For the potentially causative variants identified by colocalization analysis listed in Suppl. Table 11 (and partially discussed in the main text) it would be helpful to add the CADD scores, etc..

RESPONSE: We have added this information to Supplementary Tables 12—14 (previously Supplementary Tables 11—13).

12. [lines 180-185] The following sentence is not sufficiently clear. ”As further evidence that two distinct signals drive the daytime sleepiness and daytime napping associations, we conducted conditional analysis using GCTA COJO adjusting the daytime napping association for the lead daytime napping variant (rs2653349), and observed attenuation in the association for the lead daytime sleepiness variant in HCRTR2 (rs3122170) (P value from 4.60×10^{-18} to 4.56×10^{-3}).” If the signals are independent, conditional analysis should NOT substantially change the p-values.

RESPONSE: We agree with the reviewer that this was not sufficiently clear in the text. We now state (page 9, line 262): “Although an intronic lead variant in *HCRTR2* was previously reported in GWAS of daytime sleepiness²⁴ (rs3122170, $r^2=0.29$ with lead napping variant rs2653349), the traits in the colocalization cluster excluded the daytime sleepiness phenotype, suggesting that the newly observed napping signal is driven by a distinct causal variant in *HCRTR2* (**Supplementary Table 12**). To further explore the independence of these signals, we used GCTA COJO to perform conditional analysis adjusting the regional napping associations for the lead napping signal in *HCRTR2*. We found substantial attenuation in the association with napping for the lead daytime sleepiness variant in *HCRTR2* (rs3122170; P value from 4.60×10^{-18} to 4.56×10^{-3} after conditioning). This further suggests that the newly identified signal for daytime napping is distinct from the previously reported daytime sleepiness signal in the *HCRTR2* region.”

Reviewer #3 (Remarks to the Author):

Authors have performed gwas on day time napping in UKB and 23andME. They identified 123 loci of which 60 were replicated. Authors identify association with hypertension and waist circumference and further identify causal links. The manuscript is well written and analyses are performed carefully. I have a couple of issues:

We thank the Reviewer for their review of our manuscript.

1. The accelerometer provides an objective assessment of day time inactivity/napping. If the associations were true they should have more power to be detected with accelerometer data but what you see is only a few variants appear associated with the accelerometer based assessment. Would the authors comment on why this is?

RESPONSE: As stated by the Reviewer, the purpose of quantifying the impact of daytime napping on daytime inactivity duration from accelerometer is to partly validate the specificity of our discovered loci from self-report with an objectively determined napping behavior.

We know from prior work that genetic correlations between self-report and accelerometer-derived traits generally tend to be modest. For example, we have previously reported that the genetic correlation between self-report and accelerometer-derived **sleep duration** was $r_g = 0.43$ (reference 26).

In the case of **daytime napping**, the genetic correlation (r_g) was $r_g = 0.34$. Three possible explanations exist that may explain this modest correlation:

- a) Phenotypic differences: Daytime inactivity duration from accelerometer is the objectively derived measure most closely related to self-reported daytime napping, however important distinctions exist between these two measures. Self-report asks about daytime napping **frequency** (always, sometimes, vs. never) whereas our derived accelerometer measure is based on daytime inactivity **duration** (in minutes). With the absence of sleep diary records in the UK Biobank, it is difficult to make the distinction between daytime napping and daytime inactivity behavior from accelerometer alone (reference 68). Instead, it is more feasible to derive indicators of no movement during the day (i.e. daytime inactivity duration) without sleep diary data. Consequently, the phenotypic correlation between self-reported daytime napping **frequency** and accelerometer-derived daytime inactivity **duration** is $r = 0.17$.
- b) Sample size differences: More participants self-reported daytime napping compared to wearing the accelerometer, contributing to differences in sample size and likely statistical power: $n = 452,633$ in self-report vs. $n = 85,670$ in accelerometer.
- c) Lapsed time between measurements: The accelerometer was worn between 2.8 and 9.7 years after study baseline when daytime napping frequency was self-reported.

Despite these differences, we still observe consistency between our genetic findings from self-report with findings from accelerometer (raw values in **Supplementary Table 7**) providing important validation for our findings with an objective measure (see **figure below**).

In our Discussion, we elaborate on potential reasons for the modest replication with accelerometer data (page 21, line 755): “Our effort to partly validate the specificity of our discovered loci from self-report with an objectively determined daytime napping behavior from accelerometer was likely limited as a result of phenotypic differences between self-report and accelerometer (self-report was based on daytime napping frequency whereas accelerometer measures was based on daytime inactivity duration in the absence of sleep diaries; Pearson correlation $r^2 = 0.17$), relatively smaller sample size in the accelerometer subsample ($n = 85,670$), or lapsed time between measurements as the accelerometer was worn between 2.8 and 9.7 years after study baseline. Replication of most loci and the specific association with daytime activity duration, but not other accelerometer measures, however, support our findings.”

2. In their MR analysis, authors mention that SBP and DBP summary data are not available and therefore split the UKB into two sets. Have the authors checked with ICBP consortium. They have led the largest GWAS and summary statistics are also available upon request.

RESPONSE: We had previously attempted to obtain these summary statistics by reaching out to members of the consortium, but were unsuccessful. The only publicly available summary statistics from the ICBP consortium are the combined ICB+UKB meta-analysis, but not the ICB subsample alone, which would have precluded 2-sample MR.

As further confirmation of our results, we now perform MR analyses using the 23andMe weights for the daytime napping instrument and the combined (ICB+UKB) blood pressure meta-analysis as the outcome dataset.

We found concordant effects of daytime napping on systolic and diastolic blood pressure, although with wider confidence intervals and with a smaller magnitude of effect for diastolic blood pressure.

We now describe this sensitivity analysis in the Methods (page 38, line 1206): “As sensitivity analysis, we used daytime napping variant association statistics from the 23andMe replication

sample as the exposure, and a meta-analysis including UK Biobank and the International Consortium for Blood Pressure (n ~750,000) as the outcome⁴⁶.”

Also, we describe these findings in the Results (page 15, line 473): “In sensitivity analysis, we found a consistent effect, although attenuated in magnitude for the outcome of DBP, of genetically proxied daytime napping on higher blood pressure when using variant association statistics from 23andMe as the exposure, and blood pressure in the ICB-UKB meta-analysis⁴⁸ as the outcome (DBP: 0.08 SD units, [0.003, 1.18], $P=0.04$; SBP: 0.21 SD units, [-0.02, 0.43], $P=0.07$).”

REVIEWERS' COMMENTS<

Reviewer #1 (Remarks to the Author):

The revised manuscript has substantially improved, addressing most of the comments. I agree that these findings would be of interest for readers.

Reviewer #2 (Remarks to the Author):

there is one typo in the new sentence on page 14: "Cluster-specific olygenic score associations varied across outcomes, and included associations of cluster 1 with higher blood pressure for cluster 1, and clusters 2 and 3 with adiposity traits (Table 1)." Remove "for cluster 1".

Reviewer #1 (Remarks to the Author):

The revised manuscript has substantially improved, addressing most of the comments. I agree that these findings would be of interest for readers.

RESPONSE: We would like to thank the Reviewer for their assessment of our work.

Reviewer #2 (Remarks to the Author):

there is one typo in the new sentence on page 14: "Cluster-specific polygenic score associations varied across outcomes, and included associations of cluster 1 with higher blood pressure for cluster 1, and clusters 2 and 3 with adiposity traits (Table 1)." Remove "for cluster 1".

RESPONSE: We would like to thank the Reviewer for their careful review of our work. We have revised this sentence accordingly.